# Priors, Hierarchy, and Information Asymmetry for Skill Transfer in Reinforcement Learning

**Sasha Salter,**\* **Kristian Hartikainen,**\* **Walter Goodwin, Ingmar Posner**
University of Oxford
`{sasha,kristian,walter,hip}@oxfordrobotics.institute`

## Abstract

The ability to discover behaviours from past experience and transfer them to new tasks is a hallmark of intelligent agents acting sample-efficiently in the real world. Equipping embodied reinforcement learners with the same ability may be crucial for their successful deployment in robotics. While hierarchical and KL-regularized reinforcement learning individually hold promise here, arguably a hybrid approach could combine their respective benefits. Key to these fields is the use of information asymmetry across architectural modules to bias which skills are learnt. While asymmetry choice has a large influence on transferability, existing methods base their choice primarily on intuition in a domain-independent, potentially suboptimal, manner. In this paper, we theoretically and empirically show the crucial *expressivity-transferability* trade-off of skills across sequential tasks, controlled by information asymmetry. Given this insight, we introduce **A**ttentive **P**riors for **E**xpressive and Transferable **S**kills (APES), a hierarchical KL-regularized method, heavily benefiting from both priors and hierarchy. Unlike existing approaches, APES automates the choice of asymmetry by learning it in a data-driven, domain-dependent, way based on our *expressivity-transferability* theorems. Experiments over complex transfer domains of varying levels of extrapolation and sparsity, such as robot block stacking, demonstrate the criticality of the correct asymmetric choice, with APES drastically outperforming previous methods.

## 1 Introduction

While Reinforcement Learning (RL) algorithms recently achieved impressive feats across a range of domains (Silver et al., 2017; Mnih et al., 2015; Lillicrap et al., 2015), they remain sample inefficient (Abdolmaleki et al., 2018; Haarnoja et al., 2018b) and are therefore of limited use for real-world robotics applications. Intelligent agents during their lifetime discover and reuse skills at multiple levels of behavioural and temporal abstraction to efficiently tackle new situations. For example, in manipulation domains, beneficial abstractions could include low-level instantaneous motor primitives as well as higher-level object manipulation strategies. Endowing lifelong learning RL agents (Parisi et al., 2019) with a similar ability could be vital towards attaining comparable sample efficiency.

To this end, two paradigms have recently been introduced. KL-regularized RL (Teh et al., 2017; Galashov et al., 2019) presents an intuitive approach for automating skill reuse in multi-task learning. By regularizing policy behaviour against a learnt task-agnostic prior, common behaviours across tasks are distilled into the prior, which encourages their reuse. Concurrently, hierarchical RL also enables skill discovery (Wulfmeier et al., 2019; Merel et al., 2020; Hausman et al., 2018; Haarnoja et al., 2018a; Wulfmeier et al., 2020) by considering a two-level hierarchy in which the high-level policy is task-conditioned, whilst the low-level remains task-agnostic. The lower level of the hierarchy therefore also discovers skills that are transferable across tasks. Both hierarchy and priors offer their own skill abstraction. However, when combined, hierarchical KL-regularized RL can discover multiple abstractions. Whilst prior methods attempted this (Tirumala et al., 2019; 2020; Liu et al., 2022; Goyal et al., 2019), the transfer benefits from learning both abstractions varied drastically, with approaches like Tirumala et al. (2019) unable to yield performance gains.

In fact, successful transfer of the aforementioned hierarchical and KL-regularized methods critically depends on the correct choice of information asymmetry (IA). IA more generally refers to an asymmetric masking of information across architectural modules. This masking forces independence to, and ideally generalisation across, the masked dimensions (Galashov et al., 2019). For example, for self-driving cars, by conditioning the prior only on proprioceptive information it discovers skills

---

\*Equal contribution. Correspondence to: `sasha.salter@hotmail.com`.

independent to, and shared across, global coordinate frames. For manipulation, by not conditioning on certain object information, such as shape or weight, the robot learns generalisable grasping independent to these factors. Therefore, IA biases learnt behaviours and how they transfer across environments. Previous works predefined their IAs, which were primarily chosen on intuition and independent of domain. In addition, previously explored asymmetries were narrow (Table 1), which if sub-optimal, limit transfer benefits. We demonstrate that this indeed is the case for many methods on our domains (Galashov et al., 2019; Bagatella et al., 2022; Tirumala et al., 2019; 2020; Pertsch et al., 2021; Wulfmeier et al., 2019). A more systematic, theoretically and data driven, domain dependent, approach for choosing IA is thus required to maximally benefit from skills for transfer learning.

In this paper, we employ hierarchical KL-regularized RL to effectively transfer skills across sequential tasks. We begin by theoretically and empirically showing the crucial *expressivity-transferability* trade-off, controlled by choice of IA, of skills across sequential tasks for hierarchical KL-regularized RL. Our *expressivity-transferability* theorems state that conditioning skill modules on too little or too much information, such as the current observation or entire history, can both be detrimental for transfer, due to the discovery of skills that are either too general (e.g. motor primitives) or too specialised (e.g. non-transferable task-level). We demonstrate this by ablating over a wide range of asymmetries between the hierarchical policy and prior. We show the inefficiencies of previous methods that choose highly sub-optimal IAs for our domains, drastically limiting transfer performance. Given this insight, we introduce **APES**, 'Attentive Priors for Expressive and Transferable Skills' as a method that forgoes user intuition and automates the choice of IA in a data driven, domain dependent, manner. APES builds on our *expressivity-transferability* theorems to learn the choice of asymmetry between policy and prior. Specifically, APES conditions the prior on the entire history, allowing for *expressive* skills to be discovered, and learns a low-entropic attention-mask over the input, paying attention only where necessary, to minimise covariate shift and improve *transferability* across domains. Experiments over domains of varying levels of sparsity and extrapolation, including a complex robot block stacking one, demonstrate APES' consistent superior performance over existing methods, whilst automating IA choice and by-passing arduous IA sweeps. Further ablations show the importance of combining hierarchy and priors for discovering expressive multi-modal behaviours.

## 2 SKILL TRANSFER IN REINFORCEMENT LEARNING

We consider multi-task reinforcement learning in Partially Observable Markov Decision Processes (POMDPs), defined by $M_k = (\mathcal{S}, \mathcal{X}, \mathcal{A}, r_k, p, p_k^0, \gamma)$, with tasks $k$ sampled from $p(\mathcal{K})$. $\mathcal{S}$, $\mathcal{A}$, $\mathcal{X}$ denote observation, action, and history spaces. $p(\mathbf{x}'|\mathbf{x}, \mathbf{a}) : \mathcal{X} \times \mathcal{X} \times \mathcal{A} \to \mathbb{R}_{\geq 0}$ is the dynamics model. We denote the history of observations $\mathbf{s} \in \mathcal{S}$, actions $\mathbf{a} \in \mathcal{A}$ up to timestep $t$ as $\mathbf{x}_t = (\mathbf{s}_0, \mathbf{a}_0, \mathbf{s}_1, \mathbf{a}_1, \ldots, \mathbf{s}_t)$. Reward function $r_k : \mathcal{X} \times \mathcal{A} \times \mathcal{K} \to \mathbb{R}$ is history-, action- and task-dependent.

### 2.1 HIERARCHICAL KL-REGULARIZED REINFORCEMENT LEARNING

The typical multi-task KL-regularized RL objective (Todorov, 2007; Kappen et al., 2012; Rawlik et al., 2012; Schulman et al., 2017) takes the form:

$$\mathcal{L}(\pi, \pi_0) = \mathbb{E}_{\substack{\tau \sim p_\pi(\tau), \\ k \sim p(\mathcal{K})}} \left[ \sum_{t=0}^{\infty} \gamma^t \left( r_k(\mathbf{x}_t, \mathbf{a}_t) - \alpha_0 \mathrm{D}_{\mathrm{KL}} \left( \pi(\mathbf{a}|\mathbf{x}_t, k) \parallel \pi_0(\mathbf{a}|\mathbf{x}_t) \right) \right) \right] \quad (1)$$

where $\gamma$ is the discount factor and $\alpha_0$ weighs the individual objective terms. $\pi$ and $\pi_0$ denote the task-conditioned policy and task-agnostic prior respectively. The expectation is taken over tasks and trajectories $\tau$ from policy $\pi$ and initial observation distribution $p_k^0(\mathbf{s}_0)$, (i.e. $p_\pi(\tau)$). Summation over $t$ occurs across all episodic timesteps. When optimised with respect to $\pi$, this objective can be viewed as a trade-off between maximising rewards whilst remaining close to trajectories produced by $\pi_0$. When $\pi_0$ is learnt, it can learn shared behaviours across tasks and bias multi-task exploration (Teh et al., 2017). We consider the sequential learning paradigm, where skills are learnt from past tasks, $p_{source}(\mathcal{K})$, and leveraged while attempting the transfer set of tasks, $p_{trans}(\mathcal{K})$.

While KL-regularized RL has achieved success across various settings (Abdolmaleki et al., 2018; Teh et al., 2017; Pertsch et al., 2020; Haarnoja et al., 2018a), recently Tirumala et al. (2019) proposed a hierarchical extension where policy $\pi$ and prior $\pi_0$ are augmented with latent variables, $\pi(\mathbf{a}, \mathbf{z}|\mathbf{x}, k) = \pi^H(\mathbf{z}|\mathbf{x}, k)\pi^L(\mathbf{a}|\mathbf{z}, \mathbf{x})$ and $\pi_0(\mathbf{a}, \mathbf{z}|\mathbf{x}) = \pi_0^H(\mathbf{z}|\mathbf{x})\pi_0^L(\mathbf{a}|\mathbf{z}, \mathbf{x})$, where subscripts $H$ and $L$ denote the higher and lower hierarchical levels. This structure encourages the shared low-level policy ($\pi^L = \pi_0^L$) to discover task-agnostic behavioural primitives, whilst the high-level discovers higher-level task relevant skills. By not conditioning the high-level prior on task-id, Tirumala et al.

(2019) encourage the reuse of common high-level abstractions across tasks. They also propose the following upper bound for approximating the KL-divergence between hierarchical policy and prior:

$$D_{KL}\left(\pi(\mathbf{a}|\mathbf{x}) \,\|\, \pi_0(\mathbf{a}|\mathbf{x})\right) \leq D_{KL}\left(\pi^H(\mathbf{z}|\mathbf{x}) \,\|\, \pi_0^H(\mathbf{z}|\mathbf{x})\right) + \mathbb{E}_{\pi^H}\left[D_{KL}\left(\pi^L(\mathbf{a}|\mathbf{x},\mathbf{z}) \,\|\, \pi_0^L(\mathbf{a}|\mathbf{x},\mathbf{z})\right)\right]$$
(2)

We omit task conditioning and declaring shared modules to show this bound is agnostic to this.

## 2.2 Information Asymmetry

Information Asymmetry (IA) is a key component in both of the aforementioned approaches, promoting the discovery of behaviours that generalise. IA can be understood as the masking of information accessible by certain modules. Not conditioning on specific environment aspects forces

Table 1: Previously Explored IAs

| Paper | $\pi_0^{(H)}$ input |
|---|---|
| (Tirumala et al., 2020; Liu et al., 2022) | $\mathbf{x}_t$ |
| (Bagatella et al., 2022) | $\mathbf{a}_{t-n:t}$ |
| (Bagatella et al., 2022) | $\mathbf{x}_{t-1:t}$ |
| (Pertsch et al., 2020; 2021; Ajay et al., 2020) | $\mathbf{s}_t$ |
| (Rao et al., 2021; Tirumala et al., 2019; 2020) | $\mathbf{z}_{t-1}$ |
| (Tirumala et al., 2019; Goyal et al., 2019) | $-$ |

independence and generalisation across them (Galashov et al., 2019). In the context of (hierarchical) KL-regularized RL, the explored asymmetries between the (high-level) policy, $\pi^{(H)}$, and prior, $\pi_0^{(H)}$, have been narrow (Tirumala et al., 2019; 2020; Liu et al., 2022; Pertsch et al., 2020; 2021; Rao et al., 2021; Ajay et al., 2020; Goyal et al., 2019). Concurrent with our research, Bagatella et al. (2022) published work exploring a wider range of asymmetries, closer to those we explore. We summarise explored asymmetries in Table 1 (with $\mathbf{a}_{t-n:t}$ representing action history up to $n$ steps in the past).

Choice of information conditioning heavily influences which skills can be uncovered and how well they transfer. For example, Pertsch et al. (2020) discover observation-dependent behaviours, such as navigating corridors in maze environments, yet are unable to learn history-dependent skills, such as never traversing the same corridor twice. In contrast, Liu et al. (2022), by conditioning on history, are able to learn these behaviours. However, as we will show, in many scenarios, naïvely conditioning on entire history can be detrimental for transfer, by discovering behaviours that do not generalise favourably across history instances, between tasks. We refer to this dilemma as the *expressivity-transferability* trade-off. Crucially, all previous works predefine the choice of asymmetry, based on the practitioner's intuition, that may be sub-optimal for skill transfer. By introducing theory behind the *expressivity-transferability* of skills, we present a simple data-driven method for automating the choice of IA, by learning it, yielding transfer benefits.

## 3 Model Architecture and the Expressivity-Transferability Trade-Off

To rigorously investigate the contribution of priors, hierarchy, and information asymmetry for skill transfer, it is important to isolate each individual mechanism while enabling the recovery of previous models of interest. To this end, we present the unified architecture in Fig. 1, which introduces information gating functions (IGFs) as a means of decoupling IA from architecture. Each component has its own IGF, depicted with a colored rectangle. Every module is fed all environment information $\mathbf{x}_k = (\mathbf{x}, k)$ and distinctly chosen IGFs mask which part of the input each network has access to, thereby influencing which skills they learn. By presenting multiple priors, we enable a comparison with existing literature. With the right masking, one can recover previously investigated asymmetries (Tirumala et al., 2019; 2020; Pertsch et al., 2020; Bagatella et al., 2022;

Figure 1: Hierarchical KL-regularized architecture. The hierarchical policy modules $\pi^H$ and $\pi^L$ are regularized against their corresponding prior modules $\pi_i^H$ and $\pi_i^L$. The inputs to each module are filtered by an information gating function (IGF), depicted with colored rectangles.

Goyal et al., 2019), explore additional ones, and also express purely hierarchical (Wulfmeier et al., 2019) and KL-regularized equivalents (Galashov et al., 2019; Haarnoja et al., 2018c).

### 3.1 The Information Asymmetry Expressivity-transferability Trade-Off

While existing works investigating the role of IAs for skill transfer in hierarchical KL-regularized RL have focused on multi-task learning (Galashov et al., 2019)[1], we focus on the sequential task setting, in particular the prior's $\pi_0$ ability to handle covariate shift. In contrast to multi-task learning, in the sequential setting, there exists abrupt distributional shifts, during training, for task $p(\mathcal{K})$ and trajectory $p_\pi(\tau)$ distributions. As such, it is important that the prior handles such distributional and covariate

---

[1]Concurrent with our research Bagatella et al. (2022) also investigated various IAs for sequential transfer.

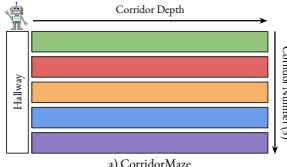 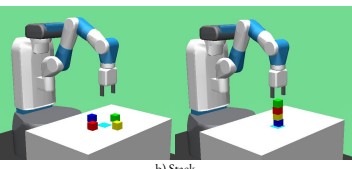

a) CorridorMaze        b) Stack

Figure 2: *Environments. a) CorridorMaze:* The agent starts in the hallway and must traverse a given sequence of corridors. The agent completes a corridor by traversing to its depth and back. *b) Stack:* The agent must stack the cubes in a given ordering over the light blue target pad.

shifts (see Theorem 3.1 for a definition). For multi-task learning, trajectory shifts are gradual and skills are continually retrained over such shifts, alleviating transfer issues. In general, IA plays a crucial role, influencing the level of covariate shift encountered by the prior during learning:

**Theorem 3.1.** *The more random variables a network depends on, the larger the covariate shift (input distributional shift, here represented by KL-divergence) encountered across sequential tasks. That is, for distributions $p$, $q$ and inputs $\mathbf{b}$, $\mathbf{c}$ such that $\mathbf{b} = (b_0, b_1, ..., b_n)$ and $\mathbf{c} \subset \mathbf{b}$:*

$$\mathrm{D}_{\mathrm{KL}}\left(p(\mathbf{b}) \parallel q(\mathbf{b})\right) \geq \mathrm{D}_{\mathrm{KL}}\left(p(\mathbf{c}) \parallel q(\mathbf{c})\right).$$

*Proof.* See Appendix B.1. □

In our case, $p$ and $q$ can be interpreted as training $p_{source}(\cdot)$ and transfer $p_{trans}(\cdot)$ distributions over network inputs, such as history $\mathbf{x}_t$ for high-level prior $\pi_0^H$. Intuitively, Theorem 3.1 states that *the more variables you condition your network on, the less likely it will transfer* due to increased covariate shift encountered between source and transfer domains, thus promoting minimal information conditioning. For example, imagine conditioning the high-level prior on either the entire history $\mathbf{x}_{0:t}$ or a subset of it $\mathbf{x}_{t-n:t}$, $n \in [0, t-1]$ (the subscript referring to the range of history values). According to Theorem 3.1, the covariate shift across sequential tasks will be smaller if we condition on a subset of the history, $\mathrm{D}_{\mathrm{KL}}\left(p_{source}(\mathbf{x}_{0:t}) \parallel p_{trans}(\mathbf{x}_{0:t})\right) \geq \mathrm{D}_{\mathrm{KL}}\left(p_{source}(\mathbf{x}_{t-n:t}) \parallel p_{trans}(\mathbf{x}_{t-n:t})\right)$. Interestingly, covariate shift is upper-bounded by trajectory shift: $\mathrm{D}_{\mathrm{KL}}\left(p_{\pi_{source}}(\tau) \parallel p_{\pi_{trans}}(\tau)\right) \geq \mathrm{D}_{\mathrm{KL}}\left(p_{\pi_{source}}(\tau_f) \parallel p_{\pi_{trans}}(\tau_f)\right)$ (using Theorem 3.1), with the right hand side representing covariate shift over network inputs $\tau_f = IGF(\tau)$, filtered trajectories (e.g. $\tau_f = \mathbf{x}_{t-n:t}$, $\tau_f \subset \tau$), and $\pi_{source}, \pi_{trans}$, source and transfer domain policies. It is therefore crucial, if possible, to minimise both trajectory and covariate shifts across domains, to benefit from previous skills. Nevertheless, the less information a prior is conditioned on, the less knowledge that can be distilled and transferred:

**Theorem 3.2.** *The more random variables a network depends on, the greater its ability to distil knowledge in the expectation (output distributional shift between network and target distribution, here represented by the expected KL-divergence). That is, for target distribution $p$ and network $q$ with outputs $\mathbf{a}$ and possible inputs $\mathbf{b}$, $\mathbf{c}$, $\mathbf{d}$, such that $\mathbf{b} = (b_0, b_1, ..., b_n)$, $\mathbf{d} \subset \mathbf{c} \subset \mathbf{b}$, $\mathbf{e} \in \mathbf{d} \oplus \mathbf{c}$:*

$$\mathbb{E}_{q(\mathbf{e}|\mathbf{d})}\left[\mathrm{D}_{\mathrm{KL}}\left(p(\mathbf{a}|\mathbf{b}) \parallel q(\mathbf{a}|\mathbf{c})\right)\right] \leq \mathrm{D}_{\mathrm{KL}}\left(p(\mathbf{a}|\mathbf{b}) \parallel q(\mathbf{a}|\mathbf{d})\right).$$

*Proof.* See Appendix B.2. □

In this particular instance, $p$ and $q$ could be interpreted as policy $\pi$ and prior $\pi_0$ distributions, $\mathbf{a}$ as action $\mathbf{a}_t$, $\mathbf{b}$ as history $\mathbf{x}_{0:t}$, and $\mathbf{c}$, $\mathbf{d}$, $\mathbf{e}$ as subsets of the history (e.g. $\mathbf{x}_{t-n:t}$, $\mathbf{x}_{t-m:t}$, $\mathbf{x}_{t-n:t-m}$ respectively, with $n > m$ and $m$ & $n \in [0, t]$), with $\mathbf{e}$ denoting the set of variables in $\mathbf{c}$ but not $\mathbf{d}$. Intuitively, Theorem 3.2 states *in the expectation, conditioning on more information improves knowledge distillation between policy and prior* (e.g. $\mathbb{E}_{\pi_0(\mathbf{x}_{t-n:t-m}|\mathbf{x}_{t-m:t})}\left[\mathrm{D}_{\mathrm{KL}}\left(\pi(\mathbf{a}_t|\mathbf{x}_{0:t}) \parallel \pi_0(\mathbf{a}_t|\mathbf{x}_{t-n:t})\right)\right] \leq \mathrm{D}_{\mathrm{KL}}\left(\pi(\mathbf{a}_t|\mathbf{x}_{0:t}) \parallel \pi_0(\mathbf{a}_t|\mathbf{x}_{t-m:t})\right)$, with $\pi_0(\mathbf{x}_{t-n:t-m}|\mathbf{x}_{t-m:t})$ the conditional distribution, induced by $\pi_0$, of history subset $\mathbf{x}_{t-n:t-m}$ given $\mathbf{x}_{t-m:t}$). Therefore, IA leads to an *expressivity-transferability* trade-off of skills (Theorems 3.1 and 3.2). Interestingly, hierarchy does not influence covariate shift and hence does not hurt transferability, but it does increase network expressivity (e.g. of the prior), enabling the distillation and transfer of rich multi-modal behaviours present in the real-world.

## 4 APES: ATTENTIVE PRIORS FOR EXPRESSIVE AND TRANSFERABLE SKILLS

While previous works chose IA on intuition (Tirumala et al., 2019; 2020; Galashov et al., 2019; Pertsch et al., 2020; Wulfmeier et al., 2019; Bagatella et al., 2022; Singh et al., 2020; Ajay et al., 2020; Liu et al., 2022) we propose learning it. Consider the information gating functions (IGFs) introduced in Section 3 and depicted in Figure 1. Existing methods can be recovered by having the IGFs perform hard attention: $IGF(\mathbf{x}_k) = \mathbf{m} \odot \mathbf{x}_k$, with $\mathbf{m} \in \{0, 1\}^{dim(\mathbf{x}_k)}$, predefined and static, and $\odot$ representing element-wize multiplication. In contrast, we propose performing soft attention with $\mathbf{m} \in [0, 1]^{dim(\mathbf{x}_k)}$ and learn $\mathbf{m}$ based on: 1) the hierarchical KL-regularized RL objective

(Equations (1) and (2)); 2) $\mathcal{L}_{IGF}(\mathbf{m}) = -\mathcal{H}(\mathbf{m})$, $\mathcal{H}$ denoting entropy (calculated by turning $\mathbf{m}$ into a probability distribution by performing Softmax over it), thereby encouraging low entropic, sparse IGFs (similar to Salter et al. (2019) applying a related technique for sim2real transfer):

$$\mathcal{L}_{\text{APES}}(\pi, \pi_0, \{\mathbf{m}_i\}^{i \in I}) = \mathbb{E}_{\substack{\tau \sim p_\pi(\tau), \\ k \sim p(\mathcal{K})}} \left[ \sum_{t=0}^{\infty} \gamma^t \left( r_k(\mathbf{x}_t, \mathbf{a}_t) - \alpha_0 \mathrm{D}_{\text{KL}}(\pi(\mathbf{a}|\mathbf{x}_k)||\pi_0(\mathbf{a}|\mathbf{x}_k)) \right) \right]$$
$$- \sum_{i \in I} \alpha_{m_i} \mathcal{H}(\mathbf{m}_i) \quad (3)$$

With $\alpha_{m_i}$ weighing the relative importance of the entropy and KL-regularized RL objectives for each attention mask $\{\mathbf{m}_i\}^{i \in I}$ for each module using self-attention (e.g. $\pi_0^H$), $I$ denoting this set. Whilst soft attention does not eliminate dimensions in the same way that hard attention does, thus losing the strict connection with Theorems 3.1 and 3.2, in practice it often leads to many 0-attention elements (Salter et al., 2020; Mott et al., 2019). $\mathbf{m}_i$ spans all history dimensions and supports intra-observation and intra-action attention. We train off-policy akin to SAC (Haarnoja et al., 2018b), sampling experiences from the replay buffer, approximating the return of the agent using Retrace (Munos et al., 2016) and double Q-learning (Hasselt, 2010) to train our critic. Refer to Appendices A and D for full training details. Exposing IGFs to all available information $\mathbf{x}_k$, we enable *expressive* skills that maximize the KL-regularized RL objective, with complex, potentially long-range, temporal dependencies (Theorem 3.2). Encouraging low-entropic masks $\mathbf{m}_i$ promotes minimal information conditioning (by limiting the IGF's channel capacity) whilst still capturing expressive behaviours. This is achieved by paying attention only where necessary to key environment aspects (Salter et al., 2022) that are crucial for decision making and hence heavily influence behaviour expressivity. Minimising the dependence on redundant information (aspects of the observation $\mathbf{s}$, action $\mathbf{a}$, or history $\mathbf{x}$ spaces that behaviours are independent to), we minimise covariate shift and improve the *transferability* of skills to downstream domains (Theorem 3.1). Consider learning the IGF of high-level prior $\pi_0^H$ for a humanoid navigation task. Low-level skills $\pi^L$ could correspond to motor-primitives, whilst the high-level prior could represent navigation skills. For navigation, joint quaternions are not relevant, but the Cartesian position is. By learning to mask parts of the observations corresponding to joints, the agent becomes invariant and robust to covariate shifts across these dimensions (unseen joint configurations). We call our method APES, 'Attentive Priors for Expressive and Transferable Skills'.

## 4.1 Training Regime and The Information Asymmetry setup

We are concerned with investigating the roles of priors, hierarchy and IA for transfer in sequential task learning, where skills learnt over past tasks $p_{source}(\mathcal{K})$ are leveraged for transfer tasks $p_{trans}(\mathcal{K})$. While one could investigate IA between hierarchical levels ($\pi^H$, $\pi^L$) as well as between policy and prior ($\pi$, $\pi_0$), we concern ourselves solely with the latter. Specifically, to keep our comparisons with existing literature fair, we condition $\pi^L$ on $\mathbf{s}_t$ and $\mathbf{z}_t$, and share it with the prior, $\pi^L = \pi_0^L$, thus enabling expressive multi-modal behaviours to be discovered with respect to $\mathbf{s}_t$ (Tirumala et al., 2019; 2020; Wulfmeier et al., 2019). In this paper, we focus on the role of IA between high-level policy $\pi^H$ and prior $\pi_0^H$ for supporting expressive and transferable high-level skills between tasks. As is common, we assume the source tasks are solved before tackling the transfer tasks. Therefore, for analysis purposes it does not matter whether we learn skills from source domain demonstrations provided by a hardcoded expert or an optimal RL agent. For simplicity, we discover skills and skill priors using variational behavioural cloning from expert policy $\pi_e$ samples:

$$\mathcal{L}_{bc}(\pi, \pi_0, \{\mathbf{m}_i\}^{i \in I}) = \sum_{j \in \{0, e\}} \mathcal{L}_{\text{APES}}(\pi, \pi_j, \{\mathbf{m}_i\}^{i \in I}) \text{ with } r_k = 0, \gamma = 1, \tau \sim p_{\pi_e}(\tau) \quad (4)$$

Equation (4) can be viewed as hierarchical KL-regularized RL in the absence of rewards and with two priors: the one we learn $\pi_0$; the other the expert $\pi_e$. See Appendix A.2 for a deeper discussion on the similarities with KL-regularized RL. We then transfer the skills and solve the transfer domains using hierarchical KL-regularized RL (as per Equation (3)). To compare the influence of distinct IAs for transfer in a controlled way, we propose the following regime: **Stage 1)** train a single hierarchical policy $\pi$ in the multi-task setup using Equation (4), but prevent gradient flow from prior $\pi_0$ to policy. Simultaneously, based on the ablation, train distinct priors (with differing IAs) on Equation (4) to imitate the policy. As such, we compare various IAs influence on skill distillation and transfer in a controlled manner, as they all distil behaviours from the same policy; **Stage 2)** freeze the shared modules ($\pi^L$, $\pi_0^H$) and train a newly instantiated $\pi^H$ on the transfer task. By freezing $\pi^L$, we assume the low-level skills from the source domains suffice for the transfer domains, often called the modularity assumption (Khetarpal et al., 2020a; Salter et al., 2022). While appearing restrictive,

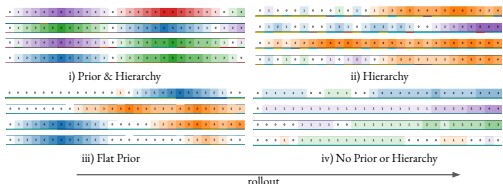

Figure 3: Skill-level exploration for *Sparse 2 corr*. 4 rollouts, episodes unrolled horizontally. Corridors are colour coded and depth within them is denoted by shade; the darker the deeper. Hallway is white. Only the full setup leads to corridor-level exploration.

the increasingly diverse the source domains are (commonly desired in settings like lifelong learning (Khetarpal et al., 2020a) and offline RL), the increasingly probable the optimal transfer policy can be obtained by recomposing the learnt skills. If this assumption does not hold, one could either fine-tune $\pi^L$ during transfer, which would require tackling the catastrophic forgetting of skills (Kirkpatrick et al., 2017), or train additional skills (by expanding $\mathbf{z}$ dimensionality for discrete $\mathbf{z}$ spaces). We leave this as future work. We also leave extending APES to sub-optimal demonstration learning as future work, potentially using advantage-weighted regression (Peng et al., 2019), rather than behavioral cloning, to learn skills. For further details refer to Appendices A and D.

## 5 EXPERIMENTS AND RESULTS

Our experiments are designed to answer the following sequential task questions: **(1)** Can we benefit from both hierarchy and priors for effective transfer? **(2)** How important is IA choice between high-level policy and prior. Does it lead to an impactful *expressivity-transferability* trade-off? In practice, how detrimental is covariate shift for transfer? **(3)** How favourably does *APES* automate the choice of IA for effective transfer? Which IAs are discovered? **(4)** How important is hierarchy for transferring expressive skills? Is hierarchy necessary? We compare against competitive skill transfer baselines, primarily Tirumala et al. (2019; 2020); Pertsch et al. (2020); Wulfmeier et al. (2019), on similar navigation and manipulation tasks for which they were originally designed and tested against.

### 5.1 ENVIRONMENTS

We evaluate on two domains: one designed for controlled investigation of core agent capabilities and the another, more practical, robotics domain (see Figure 2). Both exhibit modular behaviours whose discovery could yield transfer benefits. See Appendix C for full environmental setup details.

- **CorridorMaze.** The agent must traverse corridors in a given ordering. We collect $4*10^3$ trajectories from a scripted policy traversing any random ordering of two corridors ($p_{source}(\mathcal{K})$). For transfer ($p_{trans}(\mathcal{K})$), an inter- or extrapolated ordering must be traversed (number of sequential corridors $= \{2, 4\}$) allowing us to inspect the generalization ability of distinct priors to increasing levels of covariate shift. We also investigate the influence of covariate shift on effective transfer across reward sparsity levels: *s-sparse* (short for semi-sparse), rewarding per half-corridor completion; *sparse*, rewarding on task completion. Our transfer tasks are *sparse 2 corr* and *s-sparse 4 corr*.

- **Stack.** The agent must stack a subset of four blocks over a target pad in a given ordering. The blocks have distinct masses and only lighter blocks should be placed on heavier ones. Therefore, discovering temporal behaviours corresponding to sequential block stacking according to mass, is beneficial. We collect $17.5*10^3$ trajectories from a scripted policy, stacking any two blocks given this requirement ($p_{source}(\mathcal{K})$). The extrapolated transfer task ($p_{trans}(\mathcal{K})$), called *4 blocks*, requires all blocks be stacked according to mass. Rewards are given per individual block stacked.

### 5.2 HIERARCHY AND PRIORS FOR KNOWLEDGE TRANSFER

Table 2: Average return across 100 episodes (mean $\pm$ standard deviation for 7 random seeds). Experiments that do not (*(Hier-)RecSAC*) / do leverage prior experience, were ran for $10^6$ vs $1.5*10^5$ environment steps. *APES* employs hierarchy, priors and learns $\mathbf{m}$ for $\pi_0^H$. *APES*-{H20 - S} do not learn $\mathbf{m}$. The remainder do not employ priors.

| Approach | Paper | $\pi_0^H$ input | learn $\mathbf{m}$ | sparse 2 corr (int) | s-sparse 4 corr (ext) | 4 blocks (ext) |
|---|---|---|---|---|---|---|
| APES | - | $\mathbf{x}_{t-20:t}$ | ✓ | **0.92 ± 0.04** | **7.03 ± 0.06** | **3.36 ± 0.12** |
| APES-H20 | Tirumala et al. (2020) | $\mathbf{x}_{t-20:t}$ | ✗ | 0.15 ± 0.07 | 3.79 ± 0.14 | 1.10 ± 0.31 |
| APES-H10 | - | $\mathbf{x}_{t-10:t}$ | ✗ | 0.25 ± 0.07 | 4.12 ± 0.41 | 1.25 ± 0.19 |
| APES-H1 | Bagatella et al. (2022) | $\mathbf{x}_{t-1:t}$ | ✗ | 0.80 ± 0.02 | 6.33 ± 0.13 | **3.13 ± 0.09** |
| APES-S | Galashov et al. (2019) Pertsch et al. (2020) | $\mathbf{s}_t$ | ✗ | 0.00 ± 0.00 | 2.96 ± 0.32 | 2.05 ± 0.11 |
| APES-no_prior | Tirumala et al. (2019) | - | - | 0.00 ± 0.00 | 2.30 ± 0.11 | 0.00 ± 0.00 |
| Hier-RecSAC | Wulfmeier et al. (2019) | - | - | 0.00 ± 0.00 | 0.07 ± 0.02 | 0.01 ± 0.00 |
| RecSAC | Haarnoja et al. (2018b) | - | - | 0.00 ± 0.00 | 0.08 ± 0.01 | 0.01 ± 0.00 |
| | | Expert | | 1 | 8 | 4 |

Our full setup, *APES*, leverages hierarchy and priors for skill transfer. The high-level prior is given access to the history (as is common for POMDPs) and learns sparse self-attention $\mathbf{m}$. To investigate the importance of priors, we compare against *APES-no_prior*, a baseline from Tirumala et al. (2019), with the full *APES* setup except without a learnt prior. Comparing transfer results in Table 2, we see *APES*' drastic gains highlighting the importance of temporal high-level behavioural priors. To inspect the transfer importance of the traditional hierarchical setup (with $\pi^L(\mathbf{s}_t, \mathbf{z}_t)$), we compare *APES-no_prior* against two baselines trained solely on the transfer task. *RecSAC* represents a history-dependent *SAC* (Haarnoja et al., 2018d) and *Hier-RecSAC* a hierarchical equivalent from Wulfmeier et al. (2019). *APES-no_prior* has marginal benefits showing the importance of both hierarchy and priors for transfer. See Table 6 for a detailed explanation of all baseline and ablation setups.

## 5.3 INFORMATION ASYMMETRY FOR KNOWLEDGE TRANSFER

To investigate the importance of IA for transfer, we ablate over high-level priors with increasing levels of asymmetry (each input a subset of the previous): *APES-{H20, H10, H1, S}*, *S* denoting an observation-dependent high-level prior, *Hi* a history-dependent one, $\mathbf{x}_{t-i:t}$. Crucially, these ablations do not learn $\mathbf{m}$, unlike *APES*, our full method. Ablating history lengths is a natural dimension for POMDPs where discovering belief states by history conditioning is crucial (Thrun, 1999). *APES-H1, APES-S*

Table 3: $\mathcal{H}(\mathbf{m})$ vs $\mathrm{D_{KL}}(\pi^H || \pi_0^H)$

| Metric Domain | $\mathcal{H}(\mathbf{m})$ Corr/Stack | $\mathrm{D_{KL}}(\pi^H || \pi_0^H)$ Corr/Stack |
|---|---|---|
| APES-S | 0.70/0.70 | 0.81/0.75 |
| APES-H1 | 0.78/0.95 | 0.22/0.65 |
| APES-H10 | 1.78/1.96 | 0.12/0.49 |
| APES-H20 | 2.08/2.26 | 0.11/0.47 |
| APES | 0.26/1.20 | 0.16/0.49 |
| Max | 2.08/2.26 | 0.84/1.71 |
| Min | 0.00/0.00 | 0.00/0.00 |

are hierarchical extensions of Bagatella et al. (2022); Galashov et al. (2019) respectively, and *APES-H20* (representing entire history conditioning) is from Tirumala et al. (2020). *APES-S* is also an extension of Pertsch et al. (2020) with $\pi^L(\mathbf{s}_t, \mathbf{z}_t)$ rather than $\pi^L(\mathbf{z}_t)$. Table 2 shows the heavy IA influence, with the trend that conditioning on too little or much information limits performance. The level of influence depends on reward sparsity level: the sparser, the heavier influence, due to rewards guiding exploration less. Regardless of the transfer domain being interpolated or extrapolated, IA is influential, suggesting that IA is important over varying levels of sparsity and extrapolation.

## 5.4 THE EXPRESSIVITY-TRANSFERABILITY TRADE-OFF

To investigate whether Theorems 3.1 and 3.2 are the reason for the apparent *expressivity-transferability* trade-off seen in Table 2, we plot Figure 4 showing, on the vertical axis, the distillation loss $\mathrm{D_{KL}}\left(\pi^H \,\|\, \pi_0^H\right)$ at the end of training over $p_{source}(\mathcal{K})$, verses, on the horizontal axis, the increase in transfer performance (on $p_{trans}(\mathcal{K})$) when initialising $\pi^H$ as a task agnostic high-level policy pre-trained over $p_{source}(\mathcal{K})$ (instead

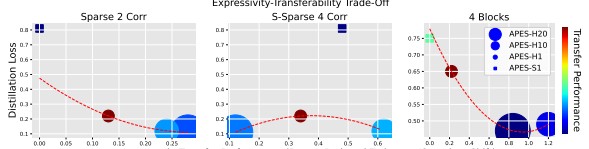

Figure 4: *Expressivity-Transferability Trade-Off*. Distillation loss vs transfer performance increase when pre-training $\pi^H$. The level of information conditioning is shown by marker size. Absolute transfer performance is shown by marker colour (red - high; blue - low). We fit a second order curve (in red) to show the general *expressivity-transferability* trend, per Theorems 3.1 and 3.2, that distilling more, by reducing IA, can hurt transfer, due to increased covariate shift. See Table 8 for detailed tabular results.

of randomly, as is default). We ran these additional pre-trained $\pi^H$ experiments to investigate whether covariate shift is the culprit for the degradation in transfer performance when conditioning the high-level prior $\pi_0^H$ on additional information. By pre-training and transferring $\pi^H$, we reduce initial trajectory shift and thus initial covariate shift between source and transfer domains (see Section 2.2). This is as no networks are reinitialized during transfer, which would usually lead to an initial shift in policy behaviour across domains. As per Theorem 3.1, we would expect a larger reduction in covariate shift for the priors that condition on more information. If covariate shift were the culprit for reduced transfer performance, we would expect a larger performance gain for those priors conditioned on more more information. Figure 4 demonstrates that is the case in general, regardless of whether the transfer domain is inter- or extra-polated. The trend is significantly less apparent for the semi-sparse domain, as here denser rewards guide learning significantly, alleviating covariate shift issues. We show results for *APES-{H20, H10, H1, S}* as each input is a subset of the previous. These relations govern Theorems 3.1 and 3.2. Figure 4 and Table 3 show that conditioning on more information improves prior expressivity, reducing distillation losses, as per Theorem 3.2. These results together with Table 2, show the impactful *expressivity-transferability* trade-off of skills, controlled by IA (as per Theorems 3.1 and 3.2), where conditioning on too little or much information limits performance.

## 5.5 APES: ATTENTIVE PRIORS FOR EXPRESSIVE AND TRANSFERABLE SKILLS

As seen in Table 2, *APES*, our full method, strongly outperforms (almost) every baseline and ablation on each transfer domain. Comparing *APES* with *APES-H20*, the most comparable approach with the prior fed the same input $(\mathbf{x}_{t-20:t})$, we observe drastic performance gains. These results demonstrate the importance of reducing covariate shift (by minimising information conditioning), whilst still supporting expressive behaviours (by exposing the prior to maximal infor-

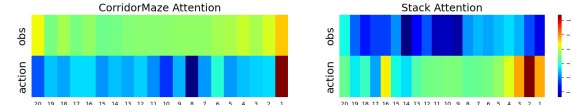

Figure 5: *APES* attention for $\pi_0^H$, plotted as $\log_{10}(\mathbf{m})$, for each domain (key on right; red and blue as high and low values). We aggregate attention to the observation/action levels, e.g. $\log_{10}(\mathbf{m}_{s_t}) = \log_{10}(\sum_{i=0}^{dim(\mathcal{S})} \mathbf{m}_{s_t}^i)$ (with $\mathbf{m}_{s_t}$ the aggregate attention for $s_t$ and $\mathbf{m}_{s_t}^i$ attention for the $i^{th}$ dimension of $s_t$). For intra-observation/action attention visualisation, reporting values per individual dimension, see Figure 7. *APES* learns sparse, domain dependent, attention, primarily focusing on recent actions.

mation), only achieved by *APES*. Table 3 shows $\mathcal{H}(\mathbf{m})$, each $\pi_0^H$ mask's entropy (a proxy for the amount of information conditioning), vs $D_{KL}(\pi^H||\pi_0^H)$ (distillation loss), reporting max/min scores across all experiments cycles. We ran 4 random seeds but omit standard deviations as they were negligible. *APES* not only attends to minimal information ($\mathcal{H}(\mathbf{m})$), but for that given level achieves a far lower distillation loss than comparable methods. This demonstrates *APES* pays attention only where necessary. We inspect *APES'* attention masks $\mathbf{m}$ in Figure 5 (aggregated at the observation and action levels). Firstly, many attention values tend to 0 (seen clearly in Figure 7) aligning *APES* closely to Theorems 3.1 and 3.2. Secondly, *APES* primarily pays attention to the recent history of actions. This is interesting, as is inline with recent concurrent work (Bagatella et al., 2022) demonstrating the effectiveness of *state-free* priors, conditioned on a history of actions, for effective generalization. Unlike Bagatella et al. (2022) that need exhaustive history lengths sweeps for effective transfer, our approach learns the length in an automated domain dependent manner. As such, our learnt history lengths are distinct for *CorridorMaze* and *Stack*. For *CorridorMaze*, some attention is paid to the most recent observation $\mathbf{s}_t$, which is unsurprising as this information is required to infer how to optimally act whilst at the end of a corridor. In Figure 7, we plot intra-observation and action attention, and note that for *Stack*, *APES* ignores various dimensions of the observation-space, further reducing covariate shift. Refer to Appendix E.1 for an in-depth analysis of *APES'* full attention maps.

## 5.6 HIERARCHY FOR EXPRESSIVITY

To further investigate whether hierarchy is necessary for effective transfer, we compare *APES-H1* with *APES-H1-flat*, which has the same setup except with a flat, non-hierarchical prior (conditioned on $\mathbf{x}_{t-1:t}$). With a flat prior, KL-regularization must occur over the raw, rather than latent, action space. There-

Table 4: Hierarchy Ablation

| Transfer task | sparse 2 corr | s-sparse 4 corr |
|---|---|---|
| APES-H1 | **0.80 ± 0.02** | **6.33 ± 0.13** |
| APES-H1-KL-a | **0.75 ± 0.07** | 5.65 ± 0.13 |
| APES-H1-flat | 0.06 ± 0.03 | 4.42 ± 0.41 |
| Expert | 1 | 8 |

fore, to adequately compare, we additionally investigate a hierarchical setup where regularization occurs only over the action-space, *APES-H1-KL-a*. Transfer results for *CorridorMaze* are shown in Table 4 (7 seeds). Comparing *APES-H1-KL-a* and *APES-H1-KL-flat*, we see the benefits of a hierarchical prior, more significant for sparser domains. Upon inspection (see Section 5.7), *APES-H1-KL-flat* is unable to solve the task as it cannot capture multi-modal behaviours (at corridor intersections). Contrasting *APES-H1* with *APES-H1-KL-a*, we see minimal benefits for latent regularization, suggesting with alternate methods for multi-modality, hierarchy may not be necessary.

## 5.7 SKILL-LEVEL EXPLORATION ANALYSIS

To gain a further understanding on the effects of hierarchy and priors, we visualise policy rollouts early on during transfer ($5 * 10^3$ steps). For *CorridorMaze* (Figure 3), with hierarchy and priors *APES* explores randomly at the corridor level. Hierarchy alone, unable to express preference over high-level skills, leads to temporally uncorrelated behaviours, unable to explore at the corridor level. The flat prior, unable to represent multi-modal behaviours, leads to suboptimal exploration at the intersection of corridors, with the agent often remaining static. Without priors nor hierarchy, exploration is further hindered, rarely traversing corridor depths. For *Stack* (Figure 6), *APES* explores at the block stacking level, alternating block orderings but primarily stacking lighter upon heavier. Hierarchy alone is unable to stack blocks with temporally uncorrelated skills, exploring at the intra-block stacking level, switching between blocks before successfully stacking, or often interacting with, any individual one.

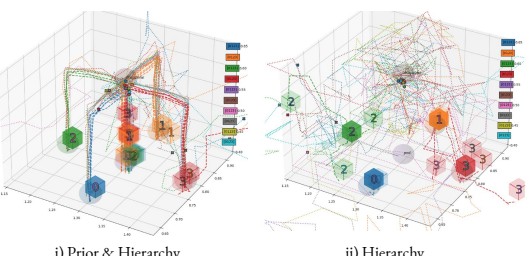

i) Prior & Hierarchy  ii) Hierarchy

Figure 6: Skill-level exploration for *4 Blocks*. End-effector and block rollouts depicted by dotted and dashed lines respectively, colour-coded per rollout. The end positions for gripper and blocks are represented by cross or cubes, respectively. Cube numbers represent their mass order with 0 the heaviest. *APES* explores at the stacking level whilst hierarchy alone is unable to stack. See Figure 10 for video rollouts.

## 6 RELATED WORK

Hierarchical frameworks have a long history (Sutton et al., 1999). The options semi-MDP literature explore hierarchy and temporal abstractions (Nachum et al., 2018; Wulfmeier et al., 2020; Igl et al., 2019; Salter et al., 2022; Kamat & Precup, 2020; Harb et al., 2018; Riemer et al., 2018). Approaches like Wulfmeier et al. (2019; 2020) use hierarchy to enforce knowledge transfer through shared modules. Gehring et al. (2021) use hierarchy to discover skills of varying expressivity levels. For lifelong learning (Parisi et al., 2019; Khetarpal et al., 2020b), where number of skills increase over time, it is unclear how well these approaches will fare, without priors to narrow skill exploration.

Priors have been used in various fields. In the context of offline-RL, Siegel et al. (2020); Wu et al. (2019) primarily use priors to tackle value overestimation (Levine et al., 2020). In the variational literature, priors have been used to guide latent-space learning (Hausman et al., 2018; Igl et al., 2019; Pertsch et al., 2020; Merel et al., 2018). Hausman et al. (2018) learn episodic skills, limiting their ability to transfer. Igl et al. (2019); Khetarpal et al. (2020a) learn options together with priors or interest functions, respectively. The primer is on-policy, limiting applicability and sample efficiency. Both predefine information conditioning of shared modules, limiting transferability. Skills and priors have been used in model-based RL to improve planning (Xie et al., 2020; Shi et al., 2022). Sikchi et al. (2022) use priors to reduce covariate shifts during planning. In contrast, APES ensures the priors themselves experience reduced shifts. In the multi-task literature, priors have been used to guide exploration (Pertsch et al., 2020; Galashov et al., 2019; Siegel et al., 2020; Pertsch et al., 2021; Teh et al., 2017), yet without hierarchy expressivity in learnt behaviours is limited. In the sequential transfer literature, priors have also been used to bias exploration (Pertsch et al., 2020; Ajay et al., 2020; Singh et al., 2020; Bagatella et al., 2022; Goyal et al., 2019; Rao et al., 2021; Liu et al., 2022), yet either do not leverage hierarchy (Pertsch et al., 2020) or condition on minimal information (Ajay et al., 2020; Singh et al., 2020; Rao et al., 2021), limiting expressivity. Unlike *APES*, Singh et al. (2020); Bagatella et al. (2022) leverage flow-based transformations to achieve multi-modality. Unlike many previous works, we consider the POMDP setting, arguably more suited for robotics, and learn the information conditioning of priors on based on our *expressivity-transferability* theorems.

Whilst most previous works rely on IA, choice is primarily motivated by intuition. For example, Igl et al. (2019); Wulfmeier et al. (2019; 2020) only employ task or goal asymmetry and Tirumala et al. (2019); Merel et al. (2020); Galashov et al. (2019) use exteroceptive asymmetry. Salter et al. (2020) investigate a way of learning asymmetry for sim2real domain adaptation, but condition **m** on observation and state. We consider exploring this direction as future work. We provide a principled investigation on the role of IA for transfer, proposing a method for automating the choice.

## 7 CONCLUSION

We employ hierarchical KL-regularized RL to efficiently transfer skills across sequential tasks, showing the effectiveness of combining hierarchy and priors. We theoretically and empirically show the crucial *expressivity-transferability* trade-off, controlled by IA choice, of skills for hierarchical KL-regularized RL. Our experiments validate the importance of this trade-off for both interpolated and extrapolated domains. Given this insight, we introduce APES, 'Attentive Priors for Expressive and Transferable Skills' automating the IA choice for the high-level prior, by learning it in a data driven, domain dependent, manner. This is achieved by feeding the entire history to the prior, capturing *expressive* behaviours, whilst encouraging its attention mask to be low entropic, minimising covariate shift and improving *transferability*. Experiments over domains of varying sparsity levels demonstrate APES' consistent superior performance over existing methods, whilst by-passing arduous IA sweeps. Ablations demonstrate the importance of hierarchy for prior expressivity, by supporting multi-modal behaviours. Future work will focus on additionally learning the IGFs between hierarchical levels.

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

# A  METHOD

## A.1  TRAINING REGIME

In this section, we algorithmically describe our training setup. We relate each training phase to the principle equations in the main paper, but note that Appendices A.2 and A.3 outline a more detailed version of these equations that were actually used. We note that during BC, we apply DAGGER (Ross et al., 2011), as per Algorithm 2, improving learning rates. For further details refer to Appendix D.

| **Algorithm 1** APES training regime |
| --- |
| 1: *# Full training and transfer regime. For BC, gradients are prevented from flowing from $\pi_0$ to $\pi$. In practice $\pi_0 = \{\pi_i\}_{i \in \{0,...,N\}}$, multiple trained priors. During transfer, $\pi^H$, is reinitialized.* |
| 2: |
| 3: *# Behavioral Cloning* |
| 4: Initialize: policy $\pi$, prior $\pi_0$, replay $R_{bc}$, DAGGER rate $r$, environment $env$ |
| 5: **for** Number of BC training steps **do** |
| 6:     $R_{bc}, env \leftarrow$ collect($\pi, R_{bc}, env$, True, $r$) |
| 7:     $\pi, \pi_0 \leftarrow$ BC_update($\pi, \pi_0, R_{bc}$) *# Eq. 4* |
| 8: **end for** |
| 9: *# Reinforcement Learning* |
| 10: Initialize: high level policy $\pi^H$, critics $Q_{k \in \{1,2\}}$, replay $R_{rl}$, transfer environment $env_t$ |
| 11: **for** Number of RL training steps **do** |
| 12:     $R_{rl}, env_t \leftarrow$ collect($\pi, R_{rl}, env_t$) |
| 13:     $\pi^H \leftarrow$ RL_policy_update($\pi, \pi_0, R_{rl}$) *# Eq. 3* |
| 14:     $Q_k \leftarrow$ RL_critic_update($Q_k, \pi, R_{rl}$) *# Eq. 9* |
| 15: **end for** |

| **Algorithm 2** collect |
| --- |
| 1: *# Collects experience from either $\pi_i$ or $\pi_e$, applying DAGGER at a given rate if instructed, and updates $R_j$, env accordingly.* |
| 2: **function** COLLECT($\pi_i$, $R_j$, $env$, $dag$=False, $r$=1) |
| 3:     $\mathbf{x} \leftarrow env$.observation() |
| 4:     $\pi_e \leftarrow env$.expert() |
| 5:     $\mathbf{a}_i \leftarrow \pi_i(\mathbf{x})$ |
| 6:     $\mathbf{a}_e \leftarrow \pi_e(\mathbf{x})$ |
| 7:     $\mathbf{a} \leftarrow$ Bernoulli($[\mathbf{a}_i, \mathbf{a}_e], [r, 1 - r]$) |
| 8:     $\mathbf{x}', r_k, env \leftarrow env$.step($\mathbf{a}$) |
| 9:     **if** $dag$ **then** |
| 10:       $\mathbf{a}_f \leftarrow \mathbf{a}_e$ |
| 11:     **else** |
| 12:       $\mathbf{a}_f \leftarrow \mathbf{a}_i$ |
| 13:     **end if** |
| 14:     $R_j \leftarrow R_j$.update($\mathbf{x}, \mathbf{a}_f, r_k, \mathbf{x}'$) |
| 15:     **return** $R_j, env$ |
| 16: **end function** |

## A.2  VARIATIONAL BEHAVIORAL CLONING AND REINFORCEMENT LEARNING

In the following section, we omit *APES*' specific information gating function objective $IGF(\mathbf{x}_k)$ for simplicity and generality. Nevertheless, it is trivial to extend the following derivations to *APES*.

Behavioral Cloning (BC) and KL-Regularized RL, when considered from the variational-inference perspective, share many similarities. These similarities become even more apparent when dealing with hierarchical models. A particularly unifying choice of objective functions for BC and RL that fit with off-policy, generative, hierarchical RL: desirable for sample efficiency, are:

$$\mathcal{L}_{bc}(\pi, \{\pi_i\}^{i \in I}) = -\sum_{i \in I} D_{KL}(\pi(\tau) \| \pi_i(\tau)), \quad \mathcal{L}_{rl}(\pi, \{\pi_i\}^{i \subset I}) = E_{\pi}(\tau)[R(\tau)] + \mathcal{L}_{bc}(\pi, \{\pi_i\}^{i \subset I})$$

$$(5)$$

$\mathcal{L}_{bc}$, corresponds to the KL-divergence between trajectories from the policy, $\pi$, and various priors, $\pi_i$. For BC, $i \in \{0, u, e\}$, denote the learnt, uniform, and expert priors. For BC, in practice, we train multiple priors in parallel: $\pi_0 = \{\pi_i\}_{i \in \{0,...,N\}}$. We leave this notation out for the remainder of this section for simplicity. When considering only the expert prior, this is the reverse KL-divergence, opposite to what is usually evaluated in the literature, (Pertsch et al., 2020). $\mathcal{L}_{rl}$, refers to a lower bound on the expected optimality of each prior $\log p_{\pi_i}(O = 1)$; $O$ denoting the event of achieving maximum return (return referred to as $R(.)$); refer to (Abdolmaleki et al., 2018), appendix B.4.3 for proof, further explanation, and necessary conditions. During transfer using RL, we do **not** have access to the expert or its demonstrations ($i \subset I := i \in \{0, u\}$).

For hierarchical policies, the KL terms are not easily evaluable. $D_{KL}(\pi(\tau) \| \pi_i(\tau)) \le \sum_t \mathbb{E}_{\pi(\tau)} \left[ D_{KL}\left( \pi^H(\mathbf{z}_t|\mathbf{x}_k) \| \pi_i^H(\mathbf{z}_t|\mathbf{x}_k) \right) + \mathbb{E}_{\pi^H(\mathbf{z}_t|\mathbf{x}_k)} \left[ D_{KL}\left( \pi^L(\mathbf{a}_t|\mathbf{x}_k, \mathbf{z}_t) \| \pi_i^L(\mathbf{a}_t|\mathbf{x}_k, \mathbf{z}_t) \right) \right] \right]$

[2] (Tirumala et al., 2019), is a commonly chosen upper bound. If sharing modules, e.g. $\pi_i^L = \pi^L$, or using non-hierarchical networks, this bound can be simplified (removing the second or first terms respectively). To make both Eq. (5) amendable to off-policy training (experience from $\{\pi_e, \pi_b\}$, for BC/RL respectively; $\pi_b$ representing behavioral policy), we introduce importance weighting (IW), removing off-policy bias at the expense of higher variance. Combining all the above with additional individual term weighting hyperparameters, $\{\beta_i^z, \beta_i^a\}$, we attain:

$$\tilde{D}_{KL}^{q(\tau)}(\pi(\tau)||\pi_i(\tau)) := \mathbb{E}_{q(\tau)}\left[\sum_t \nu^q[t] \cdot \left(\beta_i^z \cdot C_{i,h}(\mathbf{z}_t|\mathbf{x}_k) + \beta_i^a \cdot \mathbb{E}_{\pi^H(\mathbf{z}_t|\mathbf{x}_k)}\left[C_{i,l}(\mathbf{a}_t|\mathbf{x}_k,\mathbf{z}_t)\right]\right)\right]$$

$$\zeta_i^n = \frac{\mathbb{E}_{\pi^H(z_i|\mathbf{x}_i,k)}\left[\pi^L(\mathbf{a}_i|\mathbf{x}_i,\mathbf{z}_i,k)\right]}{n(\mathbf{a}_i|\mathbf{x}_i,k)}, \ \nu^n = \left[\zeta_1^n, \zeta_1^n\zeta_2^n, \dots, \prod_{i=1}^{\tau_t}\zeta_i^n\right], \ C_{\mu,\epsilon}(y) = \log\left(\frac{\pi_\epsilon^r(y)}{\pi_{\mu,\epsilon}^r(y)}\right)$$

$$-D_{KL}(\pi(\tau)||\pi_e(\tau)) \geq -\sum_{i\in\{0,u,e\}}\tilde{D}_{KL}^{\pi_e(\tau)}(\pi(\tau)||\pi_i(\tau)) \tag{6}$$

$$\mathbb{E}_{\substack{p(\mathcal{K}),\\\pi_0(\tau)}}\left[\log(O=1|\tau,k)\right] \geq \mathbb{E}_{\pi_b(\tau)}\left[\sum_t \nu^{\pi_b}[t]\cdot r_k(\mathbf{x}_t,\mathbf{a}_t)\right] - \sum_{i\in\{0,u\}}\tilde{D}_{KL}^{\pi_b(\tau)}(\pi(\tau)||\pi_i(\tau)) \tag{7}$$

Where $\tilde{D}_{KL}^{q(\tau)}(\pi(\tau)||\pi_i(\tau))$ (for $\{\beta_i^z, \beta_i^a\} = 1$) is an unbiased estimate for the aforementioned upper bound, using $q$'s experience. $\zeta_i^n$ is the IW for timestep $i$, between $\pi$ and arbitrary policy $n$. $\nu^n[t]$ is the $t^{th}$ element of $\nu^n$; the cumulative IW product at timestep $t$. Equations 6, 7 are the BC/RL lower bounds used for policy gradients. See Appendix B.4 for a derivation, and necessary conditions, of these bounds. For BC, this bounds the KL-divergence between hierarchical and expert policies, $\pi, \pi_e$. For RL, this bounds the expected optimality, for the learnt prior policy, $\pi_0$. Intuitively, maximising this particular bound, maximizes return for both policy and prior, whilst minimizing the disparity between them. Regularising against an uninformative prior, $\pi_u$, encourages highly-entropic policies, further aiding at exploration and stabilising learning (Igl et al., 2019).

In RL, IWs are commonly ignored (Lillicrap et al., 2015; Abdolmaleki et al., 2018; Haarnoja et al., 2018b), thereby considering each sample equally important. This is also convenient for BC, as IWs require the expert probability distribution: not usually provided. We did not observe benefits of using them and therefore ignore them too. We employ module sharing ($\pi_i^L = \pi^L$; unless stated otherwize), and freeze certain modules during distinct phases, and thus never employ more than 2 hyperparameters, $\beta$, at any given time, simplifying the hyperparameter optimisation. These weights balance an exploration/exploitation trade-off. We use a categorical latent space, explicitly marginalising over, rather than using sampling approximations (Jang et al., 2016). For BC, we train for 1 epoch (referring to training in the expectation once over each sample in the replay buffer).

### A.3 CRITIC LEARNING

The lower bound presented in Eq. (7) is non-differentiable due to rewards being sampled from the environment. Therefore, as is common in the RL literature (Mnih et al., 2015; Lillicrap et al., 2015), we approximate the return of policy $\pi$ with a critic, $Q$. To be sample efficient, we train in an off-policy manner with TD-learning (Sutton, 1988) using the Retrace algorithm (Munos et al., 2016) to provide a low-variance, low-bias, policy evaluation operator:

$$Q_t^{ret} := Q'(\mathbf{x}_t,\mathbf{a}_t,k) + \sum_{j=t}^\infty \epsilon_j^t\left[r_k(\mathbf{x}_j,\mathbf{a}_j) + \mathbb{E}_{\substack{\pi^H(\mathbf{z}|\mathbf{x}_{j+1},k),\\\pi^L(\mathbf{a}'|\mathbf{x}_{j+1},\mathbf{z},k)}}\left[Q'(\mathbf{x}_{j+1},\mathbf{a}',k)\right] - Q'(\mathbf{x}_j,\mathbf{a}_j,k)\right] \tag{8}$$

$$\mathcal{L}(Q) = \mathbb{E}_{\substack{p(\mathcal{K}),\\\pi_b(\tau)}}\left[(Q(\mathbf{x}_t,\mathbf{a}_t,k) - \arg\min_{Q_t^{ret}}(Q_t^{ret}))^2\right] \qquad \epsilon_j^t = \gamma^{j-t}\prod_{i=t+1}^j \zeta_i^b \tag{9}$$

---

[2]For proof refer to Appendix B.3

Where $Q_t^{\text{ret}}$ represents the policy return evaluated via Retrace. $Q'$ is the target Q-network, commonly used to stabilize critic learning (Mnih et al., 2015), and is updated periodically with the current Q values. IWs are not ignored here, and are clipped between $[0, 1]$ to prevent exploding gradients, (Munos et al., 2016). To further reduce bias and overestimates of our target, $Q_t^{\text{ret}}$, we apply the double Q-learning trick, (Hasselt, 2010), and concurrently learn two target Q-networks, $Q'$. Our critic is trained to minimize the loss in Eq. (9), which regularizes the critic against the minimum of the two targets produced by both target networks.

## B   THEORY AND DERIVATIONS

In this section we provide proofs for the theory introduced in the main paper and in Appendix A.

### B.1   THEOREM 1.

**Theorem 1.** *The more random variables a network depends on, the larger the covariate shift (input distributional shift, here represented by KL-divergence) encountered across sequential tasks. That is, for distributions p, q*

$$D_{\text{KL}}\left(p(\mathbf{b}) \| q(\mathbf{b})\right) \geq D_{\text{KL}}\left(p(\mathbf{c}) \| q(\mathbf{c})\right) \tag{10}$$
$$\text{with } \mathbf{b} = (b_0, b_1, ..., b_n) \text{ and } \mathbf{c} \subset \mathbf{b}.$$

**Proof**

$$
\begin{aligned}
D_{\text{KL}}\left(p(\mathbf{b}) \| q(\mathbf{b})\right) &= \mathbb{E}_{p(\mathbf{b})}\left[\log\left(\frac{p(\mathbf{b})}{q(\mathbf{b})}\right)\right] \\
&= \mathbb{E}_{p(\mathbf{d}|\mathbf{c}) \cdot p(\mathbf{c})}\left[\log\left(\frac{p(\mathbf{d}|\mathbf{c}) \cdot p(\mathbf{c})}{q(\mathbf{d}|\mathbf{c}) \cdot q(\mathbf{c})}\right)\right] \quad \text{with} \quad \mathbf{d} \in \mathbf{b} \oplus \mathbf{c} \\
&= \mathbb{E}_{p(\mathbf{c})}\left[\mathbb{E}_{p(\mathbf{d}|\mathbf{c})}[1] \cdot \log\left(\frac{p(\mathbf{c})}{q(\mathbf{c})}\right)\right] + \mathbb{E}_{p(\mathbf{c})}\left[\mathbb{E}_{p(\mathbf{d}|\mathbf{c})}\left[\log\left(\frac{p(\mathbf{d}|\mathbf{c})}{q(\mathbf{d}|\mathbf{c})}\right)\right]\right] \\
&= D_{\text{KL}}\left(p(\mathbf{c}) \| q(\mathbf{c})\right) + \mathbb{E}_{p(\mathbf{c})}\left[D_{\text{KL}}\left(p(\mathbf{d}|\mathbf{c}) \| q(\mathbf{d}|\mathbf{c})\right)\right] \\
&\geq D_{\text{KL}}\left(p(\mathbf{c}) \| q(\mathbf{c})\right) \quad \text{given} \quad \mathbb{E}_{p(\mathbf{c})}\left[D_{\text{KL}}\left(p(\mathbf{d}|\mathbf{c}) \| q(\mathbf{d}|\mathbf{c})\right)\right] \geq 0
\end{aligned}
\tag{11}
$$

### B.2   THEOREM 2.

**Theorem 2.** *The more random variables a network depends on, the greater its ability to distil knowledge in the expectation (output distributional shift between network and target distribution, here represented by the expected KL-divergence). That is, for target distribution p and network q with outputs $\mathbf{a}$ and possible inputs $\mathbf{b}$, $\mathbf{c}$, $\mathbf{d}$, such that $\mathbf{b} = (b_0, b_1, ..., b_n)$ and $\mathbf{d} \subset \mathbf{c} \subset \mathbf{b}$*

$$\mathbb{E}_{q(\mathbf{e}|\mathbf{d})}\left[D_{\text{KL}}\left(p(\mathbf{a}|\mathbf{b}) \| q(\mathbf{a}|\mathbf{c})\right)\right] \leq D_{\text{KL}}\left(p(\mathbf{a}|\mathbf{b}) \| q(\mathbf{a}|\mathbf{d})\right) \quad \text{with} \quad \mathbf{e} \in \mathbf{d} \oplus \mathbf{c} \tag{12}$$

**Proof**

$$
\begin{aligned}
D_{\text{KL}}\left(p(\mathbf{a}|\mathbf{b}) \| q(\mathbf{a}|\mathbf{d})\right) &= \mathbb{E}_{p(\mathbf{a}|\mathbf{b})}\left[\log\left(\frac{p(\mathbf{a}|\mathbf{b})}{q(\mathbf{a}|\mathbf{d})}\right)\right] \\
&= \mathbb{E}_{p(\mathbf{a}|\mathbf{b})}\left[\log p(\mathbf{a}|\mathbf{b}) - \log \mathbb{E}_{q(\mathbf{e}|\mathbf{d})}[q(\mathbf{a}|\mathbf{c})]\right] \quad \text{with} \quad \mathbf{e} \in \mathbf{d} \oplus \mathbf{c} \\
&\geq \mathbb{E}_{p(\mathbf{a}|\mathbf{b}) \cdot q(\mathbf{e}|\mathbf{d})}\left[\log\left(\frac{p(\mathbf{a}|\mathbf{b})}{q(\mathbf{a}|\mathbf{c})}\right)\right] \quad \text{given Jensen's Inequality} \\
&= \mathbb{E}_{q(\mathbf{e}|\mathbf{d})}\left[D_{\text{KL}}\left(p(\mathbf{a}|\mathbf{b}) \| q(\mathbf{a}|\mathbf{c})\right)\right]
\end{aligned}
\tag{13}
$$

### B.3   HIERARCHICAL KL-DIVERGENCE UPPER BOUND

All proofs in this section ignore multi-task setup for simplicity. Extending to this scenario is trivial.

**Upper Bound**

$$
\begin{aligned}
\mathrm{D}_{\mathrm{KL}}\left(\pi(\tau)\parallel \pi_i(\tau)\right) \leq \sum_t \mathbb{E}_{\pi(\tau)}[\mathrm{D}_{\mathrm{KL}}\left(\pi^H(\mathbf{z}_t|\mathbf{x}_t)\parallel \pi_i^H(\mathbf{z}_t|\mathbf{x}_t)\right) \\
+ \mathbb{E}_{\pi^H(\mathbf{z}_t|\mathbf{x}_t)}\left[\mathrm{D}_{\mathrm{KL}}\left(\pi^L(\mathbf{a}_t|\mathbf{x}_t,\mathbf{z}_t)\parallel \pi_i^L(\mathbf{a}_t|\mathbf{x}_t,\mathbf{z}_t)\right)\right]]
\end{aligned}
\tag{14}
$$

**Proof**

$$
\begin{aligned}
\mathrm{D}_{\mathrm{KL}}\left(\pi(\tau)\parallel \pi_i(\tau)\right) &= \mathbb{E}_{\pi(\tau)}\left[\log\left(\frac{\pi(\tau)}{\pi_i(\tau))}\right)\right] \\
&= \mathbb{E}_{\pi(\tau)}\left[\log\left(\frac{p(\mathbf{s}_0)\cdot \prod_t p(\mathbf{s}_{t+1}|\mathbf{x}_t,\mathbf{a}_t)\cdot \pi(\mathbf{a}_t|\mathbf{x}_t)}{p(\mathbf{s}_0)\cdot \prod_t p(\mathbf{s}_{t+1}|\mathbf{x}_t,\mathbf{a}_t)\cdot \pi_i(\mathbf{a}_t|\mathbf{x}_t)}\right)\right] \\
&= \mathbb{E}_{\pi(\tau)}\left[\log\left(\prod_t \frac{\pi(\mathbf{a}_t|\mathbf{x}_t)}{\pi_i(\mathbf{a}_t|\mathbf{x}_t)}\right)\right] \\
&= \sum_t \mathbb{E}_{\pi(\tau)}\left[\mathrm{D}_{\mathrm{KL}}\left(\pi(\mathbf{a}_t|\mathbf{x}_t)\parallel \pi_i(\mathbf{a}_t|\mathbf{x}_t)\right)\right] \\
&\leq \sum_t \mathbb{E}_{\pi(\tau)}[\mathrm{D}_{\mathrm{KL}}\left(\pi(\mathbf{a}_t|\mathbf{x}_t)\parallel \pi_i(\mathbf{a}_t|\mathbf{x}_t)\right) + \\
&\quad \mathbb{E}_{\pi(\mathbf{a}_t|\mathbf{x}_t)}\left[\mathrm{D}_{\mathrm{KL}}\left(\pi(\mathbf{z}_t|\mathbf{x}_t,\mathbf{a}_t)\parallel \pi(\mathbf{z}_t|\mathbf{x}_t,\mathbf{a}_t)\right)\right]] \\
&= \sum_t \mathbb{E}_{\pi(\tau)}\left[\mathbb{E}_{\pi(\mathbf{a}_t,\mathbf{z}_t|\mathbf{x}_t)}\left[\log\left(\frac{\pi(\mathbf{a}_t|\mathbf{x}_t)}{\pi_i(\mathbf{a}_t|\mathbf{x}_t)}\right) + \log\left(\frac{\pi(\mathbf{z}_t|\mathbf{x}_t,\mathbf{a}_t)}{\pi_i(\mathbf{z}_t|\mathbf{x}_t,\mathbf{a}_t)}\right)\right]\right] \\
&= \mathbb{E}_{\pi(\tau)}\left[\mathrm{D}_{\mathrm{KL}}\left(\pi(\mathbf{a}_t,\mathbf{z}_t|\mathbf{x}_t)\parallel \pi_i(\mathbf{a}_t,\mathbf{z}_t|\mathbf{x}_t)\right)\right] \\
&= \sum_t \mathbb{E}_{\pi(\tau)}[\mathrm{D}_{\mathrm{KL}}\left(\pi^H(\mathbf{z}_t|\mathbf{x}_t)\parallel \pi_i^H(\mathbf{z}_t|\mathbf{x}_t)\right) \\
&\quad + \mathbb{E}_{\pi^H(\mathbf{z}_t|\mathbf{x}_t)}\left[\mathrm{D}_{\mathrm{KL}}\left(\pi^L(\mathbf{a}_t|\mathbf{x}_t,\mathbf{z}_t)\parallel \pi_i^L(\mathbf{a}_t|\mathbf{x}_t,\mathbf{z}_t)\right)\right]]
\end{aligned}
\tag{15}
$$

### B.4 POLICY GRADIENT LOWER BOUNDS

#### B.4.1 IMPORTANCE WEIGHTS DERIVATION

$$
\tilde{\mathrm{D}}_{\mathrm{KL}}^{q(\tau)}(\pi(\tau)||\pi_i(\tau)) = ub(\mathrm{D}_{\mathrm{KL}}\left(\pi(\tau)\parallel \pi_i(\tau)\right))
\tag{16}
$$

For $\beta_i^z, \beta_i^a = 1$, where $ub(\mathrm{D}_{\mathrm{KL}}\left(\pi(\tau)\parallel \pi_i(\tau)\right))$ corresponds to the hierarchical upper bound introduced in Appendix A.2.

**Proof**

$$
\begin{aligned}
ub(\mathrm{D}_{\mathrm{KL}}\left(\pi(\tau)\parallel \pi_i(\tau)\right)) &= \sum_t \mathbb{E}_{\pi(\tau)}[\mathrm{D}_{\mathrm{KL}}\left(\pi^H(\mathbf{z}_t|\mathbf{x}_t)\parallel \pi_i^H(\mathbf{z}_t|\mathbf{x}_t)\right) \\
&\quad + \mathbb{E}_{\pi^H(\mathbf{z}_t|\mathbf{x}_t)}\left[\mathrm{D}_{\mathrm{KL}}\left(\pi^L(\mathbf{a}_t|\mathbf{x}_t,\mathbf{z}_t)\parallel \pi_i^L(\mathbf{a}_t|\mathbf{x}_t,\mathbf{z}_t)\right)\right]] \\
&= \sum_t \mathbb{E}_{q(\tau)\cdot\frac{\pi(\tau)}{q(\tau)}}[\mathrm{D}_{\mathrm{KL}}\left(\pi^H(\mathbf{z}_t|\mathbf{x}_t)\parallel \pi_i^H(\mathbf{z}_t|\mathbf{x}_t)\right) \\
&\quad + \mathbb{E}_{\pi^H(\mathbf{z}_t|\mathbf{x}_t)}\left[\mathrm{D}_{\mathrm{KL}}\left(\pi^L(\mathbf{a}_t|\mathbf{x}_t,\mathbf{z}_t)\parallel \pi_i^L(\mathbf{a}_t|\mathbf{x}_t,\mathbf{z}_t)\right)\right]] \\
&= \sum_t \mathbb{E}_{q(\tau)\cdot\prod_{i=0}^{t}\frac{\pi(\mathbf{a}_i|\mathbf{x}_i)}{q(\mathbf{a}_i|\mathbf{x}_i)}}[\mathrm{D}_{\mathrm{KL}}\left(\pi^H(\mathbf{z}_t|\mathbf{x}_t)\parallel \pi_i^H(\mathbf{z}_t|\mathbf{x}_t)\right) \\
&\quad + \mathbb{E}_{\pi^H(\mathbf{z}_t|\mathbf{x}_t)}\left[\mathrm{D}_{\mathrm{KL}}\left(\pi^L(\mathbf{a}_t|\mathbf{x}_t,\mathbf{z}_t)\parallel \pi_i^L(\mathbf{a}_t|\mathbf{x}_t,\mathbf{z}_t)\right)\right]] \\
&= \tilde{\mathrm{D}}_{\mathrm{KL}}^{q(\tau)}(\pi(\tau)||\pi_i(\tau))
\end{aligned}
\tag{17}
$$

### B.4.2 BEHAVIORAL CLONING UPPER BOUND

$$-\mathrm{D}_{\mathrm{KL}}(\pi(\tau)||\pi_e(\tau)) \geq - \sum_{i \in \{0,u,e\}} \tilde{\mathrm{D}}_{\mathrm{KL}}^{\pi_e(\tau)}(\pi(\tau)||\pi_i(\tau)) \tag{18}$$
$$for \quad \beta_i^z, \beta_i^a \geq 1$$

**Proof**

$$\begin{aligned}
\mathrm{D}_{\mathrm{KL}}\left(\pi(\tau) \parallel \pi_e(\tau)\right) &\leq \sum_{i \in \{0,u,e\}} \mathrm{D}_{\mathrm{KL}}\left(\pi(\tau) \parallel \pi_i(\tau)\right) \\
&\leq \sum_{i \in \{0,u,e\}} ub(\mathrm{D}_{\mathrm{KL}}\left(\pi(\tau) \parallel \pi_i(\tau)\right)) \\
&= \sum_{i \in \{0,u,e\}} \tilde{\mathrm{D}}_{\mathrm{KL}}^{q(\tau)}(\pi(\tau)||\pi_i(\tau)) \quad for \quad \beta_i^z, \beta_i^a = 1 \\
&\leq \sum_{i \in \{0,u,e\}} \tilde{\mathrm{D}}_{\mathrm{KL}}^{q(\tau)}(\pi(\tau)||\pi_i(\tau)) \quad for \quad \beta_i^z, \beta_i^a \geq 1
\end{aligned} \tag{19}$$

The last line holds true as each weighted term in $\tilde{\mathrm{D}}_{\mathrm{KL}}^{q(\tau)}(\pi(\tau)||\pi_i(\tau))$ corresponds to KL-divergences which are positive.

### B.4.3 REINFORCEMENT LEARNING UPPER BOUND

$$\mathbb{E}_{\substack{p(\mathcal{K}), \\ \pi_0(\tau)}} [\log(O=1|\tau,k)] \geq \mathbb{E}_{\pi_b(\tau)} \left[ \sum_t \nu^{\pi_b}[t] \cdot r_k(\mathbf{x}_t, \mathbf{a}_t) \right] - \sum_{i \in \{0,u\}} \tilde{\mathrm{D}}_{\mathrm{KL}}^{\pi_b(\tau)}(\pi(\tau)||\pi_i(\tau)) \tag{20}$$
$$for \quad \beta_i^z, \beta_i^a \geq 1 \quad and \quad r_k < 0$$

**Proof**

$$\begin{aligned}
\mathbb{E}_{\substack{p(\mathcal{K}), \\ \pi_0(\tau)}} [\log(O=1|\tau,k)] &\geq \mathbb{E}_{\pi_b(\tau)} \left[ \sum_t \nu^{\pi_b}[t] \cdot r_k(\mathbf{x}_t, \mathbf{a}_t) \right] - \mathrm{D}_{\mathrm{KL}}\left(\pi(\tau) \parallel \pi_i(\tau)\right) \\
&\geq \mathbb{E}_{\pi_b(\tau)} \left[ \sum_t \nu^{\pi_b}[t] \cdot r_k(\mathbf{x}_t, \mathbf{a}_t) \right] - \tilde{\mathrm{D}}_{\mathrm{KL}}^{\pi_b(\tau)}(\pi(\tau)||\pi_i(\tau)), \; for \; \beta_i^z, \beta_i^a = 1 \\
&\geq \mathbb{E}_{\pi_b(\tau)} \left[ \sum_t \nu^{\pi_b}[t] \cdot r_k(\mathbf{x}_t, \mathbf{a}_t) \right] - \tilde{\mathrm{D}}_{\mathrm{KL}}^{\pi_b(\tau)}(\pi(\tau)||\pi_i(\tau)), \; for \; \beta_i^z, \beta_i^a \geq 1 \\
&\geq \mathbb{E}_{\pi_b(\tau)} \left[ \sum_t \nu^{\pi_b}[t] \cdot r_k(\mathbf{x}_t, \mathbf{a}_t) \right] - \sum_{i \in \{0,u\}} \tilde{\mathrm{D}}_{\mathrm{KL}}^{\pi_b(\tau)}(\pi(\tau)||\pi_i(\tau)), \\
&\qquad for \quad \beta_i^z, \beta_i^a \geq 1
\end{aligned} \tag{21}$$

For line 1 proof see (Abdolmaleki et al., 2018). The final 2 lines hold due to positive KL-divergences.

## C ENVIRONMENTS

Here we cover each environment setup in detail, including the expert setup used for data collection.

### C.1 CORRIDORMAZE

Intuitively, the agent starts at the intersection of corridors, at the origin, and must traverse corridors, aligned with each dimension of the observation space, in a given ordering. This requires the agent to reach the end of the corridor (which we call half-corridor cycle), and return back to the origin, before the corridor is considered complete.

$s \in \{0, l\}^c$, $p(s_0) = \mathbf{0}^c$, $k = $ one-hot task encoding, $a \in [0, 1]$, $r_k^{\text{semi-sparse}}(\mathbf{x}_t, \mathbf{a}_t) = 1$ *if* agent has correctly completed the entire or half-corridor cycle *else* 0, $r_k^{\text{sparse}}(\mathbf{x}_t, \mathbf{a}_t) = 1$ *if* task complete *else* 0. Task is considered complete when a desired ordering of corridors have been traversed. $c = 5$ represents the number of corridors in our experiments. $l = 6$, the lengths of each corridor. Observations transition according to deterministic transition function $s_{t+1}^{j_t} = f(\mathbf{s}_t^{j_t}, \mathbf{a}_t)$. $j_t$ corresponds to the index of the current corridor that the agent is in (i.e. $j_t = 0$ if the agent is in corridor 0 at timestep $t$). $\mathbf{s}^i$ corresponds to the $i^{th}$ dimension of the observation. Observations transition incrementally or decrementally down a corridor, and given observation dimension $s^i$, if actions fall into corresponding transition action bins $\psi_{inc}, \psi_{dec}$. We define the transition function as follows:

$$f(\mathbf{s}_t^{j_t}, \mathbf{a}_t) = \begin{cases} \mathbf{s}_t^{j_t} + 1, & \text{if } \psi_{inc}^w(\mathbf{a}_t, j_t). \\ \mathbf{s}_t^{j_t} - 1, & \text{elif } \psi_{dec}^w(\mathbf{a}_t, j_t). \\ 0, & \text{otherwise.} \end{cases} \tag{22}$$

$\psi_{inc}^w(\mathbf{a}_t, j) = \text{bool}(\mathbf{a}_t \text{ in } [j/c, (j + 0.5 \cdot w)/c])$, $\psi_{dec}^w(\mathbf{a}_t, j) = \text{bool}(\mathbf{a}_t \text{ in } [j/c, (2 \cdot j - 0.5 \cdot w)/c])$. The smaller the $w$ parameter, the narrower the distribution of actions that lead to transitions. As such, $w$, together with $r_k$ controls the exploration difficulty of task $k$. We set $w = 0.9$. We constrain the observation transitions to not transition outside of the corridor boundaries. Furthermore, if the agent is at the origin, $s = \mathbf{0}^c$ (at the intersection of corridors), then the transition function is ran for all values of $j_t$, thereby allowing the agent to transition into any corridor.

### C.1.1 EXPERT SETUP

The expert samples actions uniformly within the optimal action bin range, from Eq. (22), that leads to the optimal state transition, traversing the correct corridor in the correct direction, according to the task, which corridors have been traversed, and which remain.

### C.2 STACK

This domain is adapted from the well known gym robotics FetchPickAndPlace-v0 environment (Plappert et al., 2018). The following modifications were made: 1) 3 additional blocks were introduced, with different colours, and a goal pad, 2) object spawn locations were not randomized and were instantiated equidistantly around the goal pad, see Fig. 2, 3) the number of substeps was increased from 20 to 60, as this reduced episodic lengths, 4) a transparent hollow rectangular tube was placed around the goal pad, to simplify the stacking task and prevent stacked objects from collapsing due to structural instabilities, 5) the arm was always spawned over the goal pad, see figure Fig. 2, 6) the observation space corresponded to gripper position and grasp state, as well as the object positions and relative positions with respect to the arm: velocities were omitted as access to such information may not be realistic for real robotic systems. $k = $ one-hot task encoding, $r_k^{\text{sparse}}(\mathbf{x}_t, \mathbf{a}_t) = 1$ *if* correct object has been placed on stack in correct ordering *else* 0.

### C.2.1 EXPERT SETUP

The expert is set up to stack the blocks in the given ordering. Each individual block stacking cycle consists of six segments: 1) Move the gripper position to a target location 20cm directly above the block, keeping the gripper open; 2) Vertically lower the gripper to 5mm over the block, keeping the gripper open; 3) Close the gripper until the object is grasped; 4) Vertically raise the gripper to 20cm above the initial block position, keeping the gripper closed; 5) Move the gripper to target location 20cm above the target pad, with the gripper closed; 6) Open the gripper until the object is dropped onto the target pad.

We use Mujoco's (Todorov et al., 2012) PD controller, given target relative desired gripper position, which coupled with Mujoco's inverse kinematics model, produces desired actions. We apply gains of 21 to these actions. One the target location is reached for a given stage, we proceed to the next. When opening or closing the gripper we apply actions of $0.05, -0.1$, for stages 3 and 6. For stage 3, we continue closing the gripper until there are contact forces between the gripper and cube. For stage 6, we continue opening the gripper until the block has dropped such that it is within 14cm of the target pad. To prevent fully deterministic samples, which can be problematic for behavioural cloning, we inject noise into the expert actions. Specifically, we add Gaussian noise with a diagonal covariance

Table 5: Feedforward Module, $\pi_{(0)}^{\{H,L\}}$

| hidden layers | $(512, 512)$ |
|---|---|
| hidden layer activation | relu |
| output activation | linear |

Table 6: Full experimental setup. Describes inputs to each module, level over which KL-regularization occurs, which modules are shared, and which are reused across training and transfer tasks.

| Name | learn $\mathbf{m}$ | $\pi_0^H$ | $\pi_0^L$ | $\pi^L$ | $\pi^H$ | KL level | $\pi^L = \pi_0^L$ | reused modules |
|---|---|---|---|---|---|---|---|---|
| APES | ✓ | $\mathbf{x}_{t-20:t}$ | $\mathbf{s}_t$ | $\mathbf{s}_t$ | $\mathbf{x}_k$ | $z$ | ✓ | $\pi_0^H, \pi_0^L, \pi^L$ |
| APES-H20 | ✗ | $\mathbf{x}_{t-20:t}$ | $\mathbf{s}_t$ | $\mathbf{s}_t$ | $\mathbf{x}_k$ | $z$ | ✓ | $\pi_0^H, \pi_0^L, \pi^L$ |
| APES-H10 | ✗ | $\mathbf{x}_{t-10:t}$ | $\mathbf{s}_t$ | $\mathbf{s}_t$ | $\mathbf{x}_k$ | $z$ | ✓ | $\pi_0^H, \pi_0^L, \pi^L$ |
| APES-H1 | ✗ | $\mathbf{x}_{t-1:t}$ | $\mathbf{s}_t$ | $\mathbf{s}_t$ | $\mathbf{x}_k$ | $z$ | ✓ | $\pi_0^H, \pi_0^L, \pi^L$ |
| APES-S | ✗ | $\mathbf{s}_t$ | $\mathbf{s}_t$ | $\mathbf{s}_t$ | $\mathbf{x}_k$ | $z$ | ✓ | $\pi_0^H, \pi_0^L, \pi^L$ |
| APES-no_prior | - | – | – | $\mathbf{s}_t$ | $\mathbf{x}_k$ | – | – | $\pi^L$ |
| Hier-RecSAC | - | – | – | $\mathbf{s}_t$ | $\mathbf{x}_k$ | – | – | – |
| RecSAC | - | – | – | $\mathbf{x}_k$ | – | – | – | – |
| APES-H1-KL-a | ✗ | $\mathbf{x}_{t-1:t}$ | $\mathbf{s}_t$ | $\mathbf{s}_t$ | $\mathbf{x}_k$ | $a$ | ✗ | $\pi_0^H, \pi_0^L$ |
| APES-H1-flat | ✗ | – | $\mathbf{x}_{t-1:t}$ | $\mathbf{s}_t$ | $\mathbf{x}_k$ | $a$ | ✗ | $\pi_0^L$ |

and standard deviation of $0.2$ per dimension. We do not apply noise to the gripper closing or opening actions.

## D  EXPERIMENTAL SETUP

We provide the reader with the experimental setup for all training regimes and environments below. We build off the softlearning code base (Haarnoja et al., 2018b). Algorithmic details not mentioned in the following sections are omitted as are kept constant with the original code base. For all experiments, we sample batch size number of **entire episodes** of experience during training.

### D.1  MODEL ARCHITECTURES

We continue by outlining the shared model architectures across domains and experiments. Each policy network (e.g. $\pi^H, \pi^L, \pi_0^H, \pi_0^L$) is comprised of a feedforward module outlined in Table 5. The softlearning repository that we build off (Haarnoja et al., 2018b), applies tanh activation over network outputs, where appropriate, to match the predefined output ranges of any given module. The critic is also comprised of the same feedforward module, but is not hierarchical. To handle historical inputs, we tile the inputs and flatten, to become one large input 1-dimensional array. We ensure the input always remains of fixed size by appropriately left padding zeros. For $\pi^H, \pi_0^H$ we use a categorical latent space of size 10. We found this dimensionality sufficed for expressing the diverse behaviours exhibited in our domains. Table 6 describes the setup for all the experiments in the main paper, including inputs to each module, level over which KL-regularization occurs ($z$ or $a$), which modules are shared (e.g. $\pi^L$ and $\pi_0^L$), and which modules are reused across sequential tasks. For the covariate-shift designed experiments in Table 8, we additionally reuse $\pi^H$ (or $\pi^L$ for RecSAC) across domains, and whose input is $\mathbf{x}_t$. For all the above experiments, any reused modules are not given access to task-dependent information, namely task-id ($k$) and exteroceptive information (cube locations for Stack domain). This choice ensures reused modules generalize across task instances.

### D.2  BEHAVIOURAL CLONING

For the BC setup, we use a deterministic, noisy, expert controller to create experience to learn off. We apply DAGGER (Ross et al., 2011) during data collection and training of policy $\pi$ as we found this aided at achieving a high success rate at the BC tasks. Our DAGGER setup intermittently during data collection, with a predefined rate, samples an action from $\pi$ instead of $\pi_e$, but still saves BC

target action $a_t$ as the one that would have been taken by the expert for $\mathbf{x}_k$. This setup helps mitigate covariate shift during training, between policy and expert. Noise levels were chosen to be small enough so that the expert still succeeded at the task. We trained our policies for one epoch (once over each collected data sample in the expectation). It may be possible to be more sample efficient, by increasing the ratio of gradient steps to data collection, but we did not explore this direction. The interplay we use between data collection and training over the collected experience, is akin to the RL paradigm. We build off the softlearning code base (Haarnoja et al., 2018b), so please refer to it for details regarding this interplay.

#### Table 7: Full Training Setup

(a) Behavioural Cloning Setup

| Environment | CorridorMaze | Stack |
|---|---|---|
| $\pi_{(i)}$ learning rate | $3e^{-4}$ | $3e^{-4}$ |
| $z$ categorical size | 10 | 10 |
| $\pi^H$ history-depth | 24 | 5 |
| $\beta_u^z$ | $1e^{-3}$ | $1e^{-3}$ |
| $\beta_u^a$ | $1e^{-2}$ | $1e^{-2}$ |
| $\beta_e^a$ | 1 | 1 |
| $\beta_0^z = \beta_0^a$ | 1 | 1 |
| $\alpha_m$ | $1e^{-1}$ | $1e^{-1}$ |
| DAGGER rate | 0.1 | 0.1 |
| batch size | 128 | 128 |
| episodic length | 24 | 35 |

(b) Reinforcement Learning Setup

| Environment | CorridorMaze | | Stack |
|---|---|---|---|
| Reward Type | sparse | semi-sparse | sparse |
| Transfer task | 2 corridor | 4 corridor | 4 blocks |
| $Q$ learning rate | $3e^{-6}$ | $3e^{-5}$ | $3e^{-5}$ |
| $\pi$ learning rate | $3e^{-4}$ | $3e^{-4}$ | $3e^{-4}$ |
| $\beta_0^{z/a}$ | $1e^{-2}$ | $1e^{-1}$ | $5e^{-2}$ |
| $\beta_u^{z/a}$ | $1e^{-2}$ | 0 | $5e^{-4}$ |
| $Q$ update rate | $6e^{-4}$ | $6e^{-4}$ | $6e^{-4}$ |
| Retrace $\lambda$ | 0.99 | 0.99 | 0.99 |
| batch size | 128 | 128 | 128 |
| episodic length | 30 | 60 | 65 |

Refer to Table 7a for BC algorithmic details. It is important to note here that, although we report five $\beta$ hyper-parameter values, there are only two degrees of freedom. As we stop gradients flowing from $\pi_0$ to $\pi$, choice of $\beta_0$ is unimportant (as long as it is not 0) as it does not influence the interplay between gradients from individual loss terms. We set these values to 1. $\beta_e^a$'s absolute value is also unimportant, and only its relative value compared to $\beta_u^z$ and $\beta_u^a$ matters. We also set $\beta_e^a$ to 1. For the remaining two hyper-parameters, $\beta_u^z, \beta_u^a$, we performed a hyper-parameter sweep over three orders of magnitude, three values across each dimension, to obtain the reported optimal values. In practice $\pi_0 = \{\pi_i\}_{i \in \{0,...,N\}}$, multiple trained priors each sharing the same $\beta_0$ hyper-parameters. For $\alpha_m$, denoting the hyper-paramter in Equation (3) weighing the relative contribution of the $\pi_0^H$'s self-attention entropy objective $IGF(\mathbf{x}_k)$ with relation to the remainder of the RL/BC objectives, we performed a sweep over three ordered of magnitude ($1e^0, 1e^{-1}, 1e^{-2}$). This sweep was ran independent of all the other sweeps, using the optimal setup for all other hyper-parameters. We chose the hyper-parameter with lowest $D_{KL}(\pi^H || \pi_0^H)$, $\mathcal{H}(\mathbf{m})$ combination. Four seeds were ran, as for all experiments. We observed very small variation in learning across the seeds, and used the best performing seed to bootstrap off for transfer. We separately also performed a hyper-parameter sweep over $\pi$ learning rate, in the same way as before. We did not perform a sweep for batch size. We found for both BC and RL setups, that conditioning on entire history for $\pi^H$ was not always necessary, and sometimes hurt performance. We state the history lengths used for $\pi^H$ for BC in Table 7a. This value was also used for both $\pi^H$ and $Q$ for the RL setup.

We prevent gradient flow from $\pi_0$ to $\pi$, to ensure as fair a comparison between ablations as possible: each prior distils knowledge from the same, high performing, policy $\pi$ and dataset. If we simultaneously trained multiple $\pi$ and $\pi_0$ pairs (for each distinct prior), it is possible that different learnt priors would influence the quality of each policy $\pi$ which knowledge is distilled off. In this paper, we are not interested in investigating how priors affect $\pi$ during BC, but instead how priors influence what knowledge can be distilled and transferred. We observed prior KL-distillation loss convergence across tasks and seeds, ensuring a fair comparison.

### D.3 REINFORCEMENT LEARNING

During this stage of training we freeze the prior and low-level policy (if applicable, depending on the ablation). In general, any reused modules across sequential tasks are frozen (apart from $\pi^H$ for the covariate shift experiments in Table 8). Any modules that are not shared (such as $\pi^H$ for most

experiments), are initialized randomly across tasks. The RL setup is akin to the softlearning repository (Haarnoja et al., 2018b) that we build off. We note any changes in Table 7b. We regularize against the latent or action level for, depending on the ablation, whether or not our models are hierarchical, share low level policies, or use pre-trained modules (low-level policy and prior). Therefore, we only ever regularize against, at most, two $\beta$ hyper-parameters. Hyper-parameter sweeps are performed in the same way as previously. We did not sweep over *Retrace* $\lambda$, *batch size*, or *episodic length*. For Retrace, we clip the importance weights between $[0, 1]$, like in Munos et al. (2016), and perform $\lambda$ returns rather than n-step. We found the Retrace operator important for sample-efficient learning.

# E  FINAL POLICY ROLLOUTS AND ANALYSIS

## E.1  ATTENTION ANALYSIS

We plot the full attention maps for *APES* in Figure 7, including intra- observation and action attention. For *CorridorMaze*, attention is primarily paid to the most recent action $\mathbf{a}_t$. For most observations in the environment (excluding end of the corridor or corridor intersection observations), conditioning on the previous action suffices to infer the

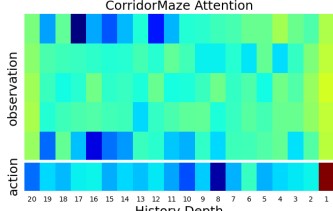 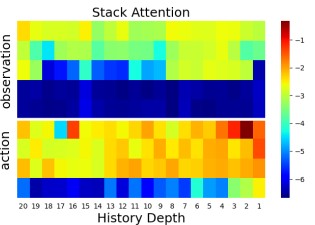

Figure 7: *APES* attention for $\pi_0^H$, plotted as $\log_{10}(\mathbf{m})$, for each domain (key on right; red and blue as high and low values). *APES* learns sparse, domain dependent, attention.

optimal next action (e.g. continue traversing the depths of a corridor). Therefore, it is understandable that *APES* has learnt such an attention mechanism. For the remainder of the environment observations (such as corridor ends), conditioning on (a history of) observations is necessary to infer optimal behaviour. As such *APES* pays some attention to observations. For *Stack*, attention is primarily paid to a short recent history of actions, with the weights decaying further into the past. Interestingly, attention over actions corresponding to opening/closing the gripper (the bottom row of Figure 7) decay a lot quicker, suggesting that this information is redundant. This makes sense, as there exists strong correlation between gripper actions at successive time-steps, but this correlation decays very quickly. Additionally, *APES* does not pay attention to observations corresponding to gripper position (the final 3 observation rows in Figure 7), as this can be inferred from the remainder of the observation-space as well as the recent history of gripper actions.

## E.2  COVARIATE SHIFT ANALYSIS

In Table 8 we report *additional return* (over a randomly initialised $\pi^H$) achieved by pre-training (over $p_{source}(\mathcal{K})$) and transferring a task-agnostic high-level policy $\pi^H$ during transfer ($p_{trans}(\mathcal{K})$). For experiments that are not hierarchical we pre-train an equivalent non-hierarchical agent. Theorem 3.1 suggests we would expect a larger improvement in transfer performance for priors that condition on more information. Table 8 confirms this trend demonstrating the importance of prior covariate shift in transferability of behaviours. This trend is less apparent for the semi-sparse domain. Additionally, for the interpolated transfer task (2 corridor),

Table 8: *Covariate Shift Analysis*. In general, reduced IA benefits more from reduced covariate shift. Sparse domains suffer more from shift, seen by clearer IA covariate trends.

|  | CorridorMaze | | Stack |
|---|---|---|---|
|  | interpolate | extrapolate | extrapolate |
|  | sparse | semi-sparse | sparse |
| Approach | 2 corridor | 4 corridor | 4 blocks |
| APES-H20 | $\mathbf{0.28 \pm 0.03}$ | $0.12 \pm 0.22$ | $0.84 \pm 0.06$ |
| APES-H10 | $0.24 \pm 0.07$ | $\mathbf{0.62 \pm 0.30}$ | $\mathbf{1.20 \pm 0.31}$ |
| APES-H1 | $0.13 \pm 0.02$ | $0.34 \pm 0.32$ | $0.22 \pm 0.24$ |
| APES-S | $0.00 \pm 0.00$ | $0.48 \pm 0.21$ | $0.00 \pm 0.17$ |
| APES-no_prior | $0.00 \pm 0.00$ | $0.09 \pm 0.08$ | $0.49 \pm 0.26$ |
| Hier-RecSAC | $0.00 \pm 0.00$ | $0.02 \pm 0.03$ | $0.00 \pm 0.01$ |
| RecSAC | $0.00 \pm 0.00$ | $0.27 \pm 0.13$ | $0.19 \pm 0.05$ |
| Expert | 0 | 0 | 0 |

the solution is entirely in the support of the training set of tasks. Naïvely, one would expect pre-training to fully recover lost performance and match the most performant method. However, this is not the case as the critic, trained solely on the transfer task, quickly encourages sub-optimal out-of-distribution behaviours.

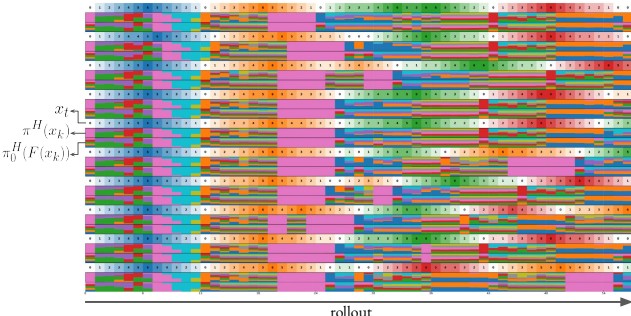

Figure 8: *CorridorMaze 4 corridor*. 10 final policy rollouts (episodes vertically, rollouts horizontally) for most performant method. Task: (Blue, Orange, Green, Red) corridors. Policy distribution over categorical latent space for $\pi^H$ and $\pi_0^H$ plotted (between policy rollouts, denoted by $x_t$) as vertical histograms, with colour and width denoting category and probability.

### E.3 FINAL POLICY ROLLOUTS

In this section, we show final policy performance (in terms of episodic rollouts) for *APES-H1*, across each transfer domain. We additionally display the categorical probability distributions for $\pi^H(\mathbf{x}_k)$ and $\pi_0^H(F(\mathbf{x}_k))$ across each rollout to analyse the behaviour of each. $F(.)$ denotes the chosen information gating function for the prior (referred to as $IGF(.)$ in the main text). For *CorridorMaze 2 and 4 corridor*, seen in Figs. 8 and 9(a), we see that the full method successfully solves each respective task, correctly traversing the correct ordering of corridors. The categorical distributions for these domains remain relatively entropic. In general, latent categories cluster into those that lead the agent deeper down a corridor, and those that return the agent to the hallway. Policy and prior align their categorical distributions in general, as expected. Interestingly, however, the two categorical distributions deviate the most from each-other at the hallway, the bottleneck state (Sutton et al., 1999), where prior multimodality (for hierarchical $\pi_0$) exists most (e.g. which corridor to traverse next). In this setting, the policy needs to deviate from the multimodal prior, and traverse only the optimal next corridor. We also observe, for the hallway, that the prior allocates one category to each of the five corridors. Such behaviour would not be possible with a flat prior.

Fig. 9(b) plots the same information for *Stack 4 blocks*. APES-H1 successfully solves the transfer task, stacking all blocks according to their masses. Similar categorical latent-space trends exist for this domain as the previous. Most noteworthy is the behaviour of both policy and prior at the bottleneck state, the location above the block stack, where blocks are placed. This location is visited five times within the episode: at the start $\mathbf{s}_0$, and four more times upon each stacked block. Interestingly, for this state, the prior becomes increasingly less entropic upon each successive visit. This suggests that the prior has learnt that the number of feasible high-level actions (corresponding to which block to stack next), reduces upon each visit, as there remains fewer lighter blocks to stack. It is also interesting that for $\mathbf{s}_0$, the red categorical value is more favoured than the rest. Here, the red categorical value corresponds to moving towards cube $0$, the heaviest cube. This behaviour is as expected, as during BC, this cube was stacked first more often than the others, given its mass. For this domain, akin to *CorridorMaze*, the policy deviates most from the prior at the bottleneck state, as here it needs to behave deterministically (regrading which block to stack next).

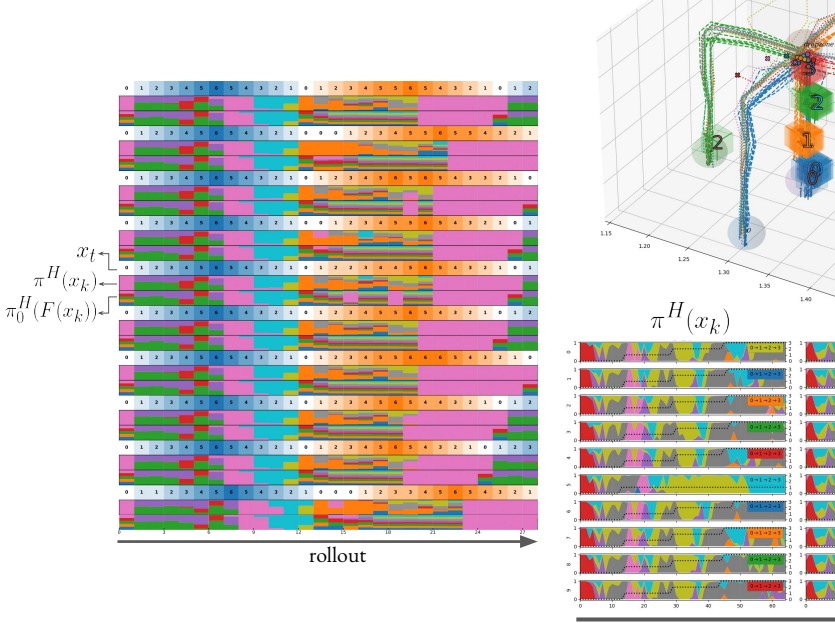

((a)) *CorridorMaze 2 corridor.* 10 final policy rollouts (episodes vertically, rollouts horizontally) for most performant method. Task: (Blue, Orange) corridors. Policy distribution over categorical latent space for $\pi^H$ and $\pi_0^H$ plotted (between policy rollouts, denoted by $x_t$) as vertical histograms, with colour and width denoting category and probability. Colour here does not correlate to corridor.

((b)) *Stack 4 blocks. Top)* Policy rollouts for most performant method. Task: stack blocks in order $(0, 1, 2, 3)$. *Bottom)* policy distributions akin to Fig. 9(a). Horizontal dashed lines in bottom plots refer to current sub-task (block stack), vertically transitioning upon each completion.

Figure 9: Final transfer performance for most performant method. Tasks are solved. Displaying latent distribution for both policy and prior. For both domains, policy deviates most from prior at the bottleneck state (hallway/stacking zone, for CorridorMaze/Stack), where prior multimodality exists (e.g. which corridor/block to stack next), but where determinism is required for the task at hand.

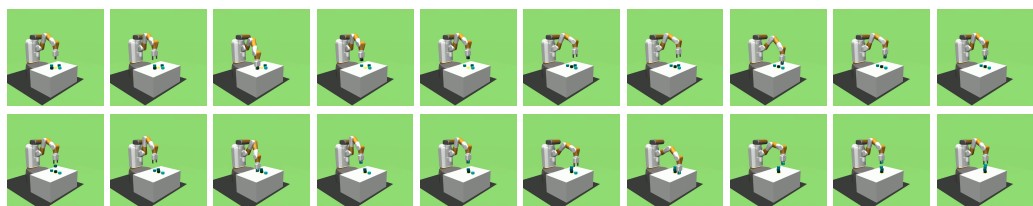

Figure 10: *Typical policy rollout, early during training, for APES on Stack 4 blocks.* We plot snapshots of a single typical policy rollout early during training. Video snapshots unroll left-to-right, top-to-bottom. *APES* explores at the individual block stacking level.

