# OpenReview forum: "Priors, Hierarchy, and Information Asymmetry for Skill Transfer in Reinforcement Learning"
_ICLR.cc/2023/Conference — ICLR 2023 poster_

### Official Review · Reviewer_ia3Q · 2022-10-24

**Confidence:** 4
**Correctness:** 3
**Technical Novelty And Significance:** 3
**Empirical Novelty And Significance:** 3
**Recommendation:** 8

**Clarity, Quality, Novelty And Reproducibility:**

The contributions are novel to the best of my understanding. The paper is clear, easy to follow, and of high quality.

**Strength And Weaknesses:**

Strengths

1. I found the unifying perspective of Information Asymmetry to be very interesting in connecting skill-learning algorithms that condition on different aspects of the observation space, and in analyzing them under a common umbrella.

2. The experimental settings of maze navigation and block stacking are insightful because the skills in these case are intuitive. Also, the tasks are non-trivial to solve which makes the comparisons with baselines clear - in that the baselines are not able to solve some of the tasks which the proposed algorithm is able to.

Weakness

1. The theoretical sections are a bit hard to follow through. In particular it is not clear whether the expressivity-transferrability tradeoff emerges due to the specific sequential nature of the problem being considered and whether the tradeoff would still exist in the usual multi-task style setup described in section 3.1

2. It is unclear what Figure 6 is trying to show: it seems like skill-level exploration, but without the corresponding visualizations of robot behavior it is unclear whether these diverse trajectories are "meaningfully diverse"

3. Some related works in model-based skill-transfer are not dicusssed (see A and B below)

A) Xie, Kevin, Homanga Bharadhwaj, Danijar Hafner, Animesh Garg, and Florian Shkurti. "Latent Skill Planning for Exploration and Transfer." In International Conference on Learning Representations. 2020.

B) Shi, Lucy Xiaoyang, Joseph J. Lim, and Youngwoon Lee. "Skill-based model-based reinforcement learning." arXiv preprint arXiv:2207.07560 (2022).

**Summary Of The Paper:**

This paper targets the problem of skill-transfer in reinforcement learning by focusing on learning information asymmetry from data. The key contribution is formalizing the notion of information asymmetry as a unifying perspective on development of skills autonomously, and providing a neat knob that can be turned to tradeoff expressivity of skills with transferability. Experiments on simulated robotics tasks like block stacking and maze navigation show the utility of the proposed framework in skill emergence and transfer.

**Summary Of The Review:**

My main concerns with the paper are minor and relate to proper explanation of some theoretical and empirical results, and some discussions of related works. These can be easily incorporated in the revisions, and as such I am recommending that the paper be accepted.

---

> ### Author Response · Authors · 2022-11-12
> **Reviewer ia3Q Rebuttal**
>
> **The theoretical section is a little hard to follow. It is unclear if the expressivity-transferability trade-off only exists for sequential task setting. Does it still exist for multi-task settings?**
>
> We thank the reviewer for this important clarification question. The expressivity-transferability trade-off does exist in other settings, such as multi-task learning. In the context of multi-task learning, Galashov 2019, Tirumala 2020, and Bagatella 2022 all demonstrate such a trade-off, through IA ablations, even though this was not their primary intention. We explain these shared trends via Theorems 3.1 and 3.2. Our theorems are agnostic to the transfer learning setting, as p() and q() can represent distinct tasks for sequential-task or multi-task learning. Nevertheless, concurrent to our work, Bagatella 2022 empirically demonstrates that the expressivity-transferability trade-off is far more drastic for the primer (see Figure 4 in their paper). The main distinction between sequential-task and multi-task learning is that for the primer, the prior is typically frozen during transfer learning (to prevent catastrophic forgetting). As such, it is crucial that it generalizes in a zero-shot fashion, with covariate shift playing a critical role. In the multi-task setting, if the prior does not generalize favorably, the agent collects new experiences and re-trains the prior over them, thus minimizing issues related to covariate-shift. We have modified Section 3.1 to clarify this point.
>
> **It is unclear what Fig 6 is trying to show. Skill-level exploration? Without visualizing the robot, it is unclear if behaviors are meaningfully diverse.**
>
> Fig 6 is trying to show the agent’s ability to transfer skills related to block stacking (especially placing lighter upon heavier blocks). In order to prevent the visualization from becoming too cluttered, we only plot the robot’s end effector (gripper) trajectory as a dotted line (color-coded per rollout), the trajectory of each block as a dashed line (color-coded per block), as well as the block’s final position, depicted as a cube (with the color and number depicting the specific block). Lower block numbers represent heavier blocks (i.e. 0 is the heaviest, and 3 is the lightest). We plot multiple rollouts. We believe this information suffices for showing what sort of skills are transferred. Notably, for APES, not only are blocks stacked at the end of each episode (seen by stacked cubes in Fig 6 (i)), but additionally, heavier blocks tend to be stacked first (seen with smaller numbers typically occurring lower down the stack). In contrast, hierarchy alone does not result in stacked blocks nor significant block manipulation (seen in Fig 6 (ii) as most cubes remain in their instantiation positions at the end of the episode). We have modified the descriptions and titles of Sections 5.7, Fig 3 and Fig 6 to try and make this clearer.
>
> We agree with the reviewer that this figure can be hard to parse. We will therefore also provide videos in the camera ready presentation, a link to these videos in the paper, and video snapshots in the Appendix. We hope this addresses the reviewer’s concerns.
>
> **Some model-based skill transfer works are not discussed or referenced.**
>
> We thank the reviewer for pointing out these relevant references. In this paper, we concern ourselves with model-free approaches. Model-based skill approaches hold promise for further increasing sample efficiency, by using both skills and transition models during transfer. If both skills and transition models are frozen during transfer, then it is crucial that both generalize in a zero-shot fashion, for similar reasons mentioned earlier (Theorems 3.1 and 3.2). Skill priors have been used in model-based RL to tackle covariate shift during planning (see [1]), by ensuring plans remain “close” to state-action tuples that the transition-model has been trained on and hence accurate over. APES could be used in this context to ensure that the skill priors themselves generalize. We have added these references and discussion points to the related works section.
>
> [1] Sikchi, Harshit, Wenxuan Zhou, and David Held. "Learning off-policy with online planning." Conference on Robot Learning. PMLR, 2022.

---

> > ### Comment · Reviewer_ia3Q · 2022-11-19
> > **Thanks for the response**
> >
> > Thank you for your detailed responses. My concerns are largely clear. I would keep my current rating of accept.

---

### Official Review · Reviewer_pBgA · 2022-10-25

**Confidence:** 4
**Correctness:** 2
**Technical Novelty And Significance:** 3
**Empirical Novelty And Significance:** 3
**Recommendation:** 5

**Clarity, Quality, Novelty And Reproducibility:**

The paper could gain a lot in terms of clarity. The idea of a expressivity-transferability trade-off is mentioned many times, before we finally understand what the authors are referring to. The theorems present the idea of covariate shift and combine it with the idea of transferability. I am not convinced that the two are equal, as transferability may depend on many more things than covariate shift and may be reflected in different difficulties that the agent has to overcome. The authors give the example of self driving cars to motivate information asymmetry, but this is only one example. How would IA be considered when transferring between two opponents in a game of Go? Is covariate shift the right way to look at transferability here? And how would covariate shift be related to IA in this case? Those are important nuances that the paper does not consider: it is written as though we assume that IA is necessary. Is the IA implemented solely through the masking in equation 9? It seems like there would be many works in the literature that use similar ideas, but are not compared to or clearly referred. Similarly, the idea of an initiation set and more recently interest function in HRL is an important parallel idea that is not mentioned.

In the experiments, why do start from expert trajectories? It seems like the proposed approach is straightforward and could be implemented in a straightforward setting, so that we clearly understand the benefits. For example, the proposed method is compared with methods that do not leverage expert trajectories. Even if they are trained longer, this seems quite unfair. Moreover, the number of seeds is only 4, which only brings additional questions as to what we can get out of the experiments.

**Strength And Weaknesses:**

# Strengths
- The paper tackles an important problem by combining two existing approaches in an interesting fashion
- The paper compares with a large set of baselines
- The theorems presented could be of interest to the community

# Weaknesses
- The claims do not seem necessarily true. In particular, the authors claim that successful transfer relies on information asymmetry, which may or may not be true, depending on the method one considers for transfer learning.
- The relation between the theorems and the ability to transfer is not necessarily direct. Some nuances would be important here
- The empirical setup is quite complex and the number of seeds is very low, bringing into question the reliability of the results.

**Summary Of The Paper:**

The paper tackles the problem of efficient transfer by combining two methods in the literature: hierarchical reinforcement learning (HRL) and KL-regularized RL. The paper argues that a key element to the transferability of the policies is by considering information asymmetry between them. To achieve this goal, the authors propose to employ soft attention on the inputs and learn this attention mask through an entropy objective on its weights (equation 9). To motivate information asymmetry, the authors two theorems on covariate shift and expectation distillation. The paper performs experiments on two domains, one a toy task and a harder stacking task, and report improved performance compared to other baselines.

**Summary Of The Review:**

The paper proposes an interesting combination between HRL and KL-regularized RL and motivate the idea of information asymmetry as a way to better transfer across tasks. The paper relates the idea of information asymmetry to transferability, which is not always true. Moreover, transferability is related to covariate, which once again is not always true. More nuances are needed. For the experiments, the setting seems more complex than it should and is only evaluated with a few seeds.

---

> ### Author Response · Authors · 2022-11-12
> **Reviewer pBgA Rebuttal (Part 1)**
>
> **The claim that IA is important for successful transfer may not be true, and depends on the method for transfer.**
>
> The statements made in this paper are in the context of hierarchical (and/or) KL-regularized RL for skill acquisition and transfer. With the hope of making this context more explicit and clear, we have made several modifications to the Introduction, Conclusion, and Section 3.1.
>
> **The empirical setup is quite complex and the number of seeds are low bringing into question the reliability of results.**
>
> We run a similar number of seeds as most of the published related works in this area (such as our baselines; Persch 2020 - 3 seeds, Galashov 2019 - 5, Tirumala 2020 - (5 for pretraining, 2 for transfer), Wulfmeier 2019 - 3, Haarnoja 2018 - 5).
>
> Given the same unifying framework to train all methods, Fig 1 and Eq 3 and 4, we consider our analysis valid, by contrasting in a controlled manner different dimensions of variations: IA, hierarchy vs no hierarchy, priors vs no priors.
>
> **The expressivity-transferability trade-off is mentioned many times before we are explained what it is.**
>
> We thank the reviewer for pointing this out. We have added sentences in the Introduction and Section 2.3 to try and address this. Does the reviewer feel this is clearer now?
>
> **The authors present only 1 intuitive example of IAs influence on transferability (cars).**
>
> We would like to highlight to the reviewer that in the original manuscript, we provide further examples in Sections 2.3 and 4.0. Nevertheless, we have added one more example in the introduction to help readers grasp the concept of IA early on in the paper.
>
> **How would IA be considered in the two player game of Go? Does it influence transferability here? What is its relation to covariate shift here?**
>
> We consider continuous control, single player domains and compare against baselines developed for the same problem setup. Nevertheless, it would be interesting to explore IA and APES in 2 (or more) player games, but this is outside the scope of this work.
>
> In the context of Go, priors could play the role of shared strategies used against different opponents. Like the domains in our paper, Go could be considered a POMDP, as we are unaware of the opponent’s policy. By conditioning our prior on all historical information, we are able to discover sophisticated (i.e. expressive) strategies that adequately model and counteract the opponent’s belief state and strategy. Nevertheless, conditioning on entire history could be detrimental for transfer, by discovering strategies that overfit to previously observed history instances (e.g. specific opponents and their strategy instantiations) and that do not generalize across them (due to high history covariate shift encountered between games and opponents). By conditioning the prior on minimal history information, through APES, we minimize covariate shift, prevent overfitting to history instantiations, and discover strategies that generalize across opponents and specific game/opponent strategy instantiations.
>
> **These nuances about IA should be clarified. We are made to believe IA is necessary for transfer.**
>
> One of the main takeaways of Galashov 2019, Tirumala 2019, 2020, Bagatella 2022, our work, and several other related works, is that while IA might not be absolutely necessary for transfer in hierarchical and KL-regularized RL, it does bring considerable benefits, both through theory and empirical studies.
>
> **Is IA implemented solely through the masking in Eq 3 (reviewer says Eq 9)?**
>
> Yes it is. To clarify, m is the attention mask over all observation-action dimensions in the history, controls IA, and is trained with Eq 3. We have modified the writing in Section 4 to clarify this, and have explicitly conditioned Eq 3 and 4 on m to highlight that it is trained via these equations.
>
> **It seems there would be many works that use similar ideas yet are not compared against or referred to.**
>
> We provide a broad comparison of recent peer reviewed baselines (e.g. Pertsch at Conference of Robot Learning 2020, Galashov at the International Conference of Learning Representations 2019, Wulfmeier at Robots Science and Systems 2020, Tirumala at the Journal of Machine Learning Research 2022). In the related works and IA sections we also highlight other works. To the best of our knowledge, we are the first approach to learn IA for skill priors. May the reviewer share the other works that they are referring to?

---

> > ### Author Response · Authors · 2022-11-12
> > **Reviewer pBgA Rebuttal (Part 2)**
> >
> > **Initiation sets and interest functions (for Options frameworks) are not mentioned.**
> >
> > Applying APES to the options framework would be an interesting extension. However, in this work we limit ourselves to KL-regularized and hierarchical frameworks. Interest functions play a similar role to option priors, like Igl 2019. Both predefine the information conditioning of the priors or interest functions, limiting transferability. APES could be extended here as a method for learning the conditioning. We have added a section to related works mentioning interest functions and initiation sets.
> >
> > **Why do we start with expert trajectories? Usual transfer setting seems simple to run and would help understanding the benefits.**
> >
> > We investigate how hierarchy, priors, and IA influence distillation and skill transfer across sequential tasks. As is common in sequential task learning, we assume the source tasks are solved before moving onto the transfer tasks. Therefore, for analysis purposes, it does not matter whether the source tasks are solved, and demonstrations are provided, by a hardcoded expert or fully optimized RL agent. In-fact, many skill transfer methods, such as Pertsch 2021 and our baseline Pertsch 2020, also assume “expert” source task trajectories for skill discovery. If one wished to learn skills from sub-optimal offline trajectories, such as those from the replay buffer of a lifelong learning sequential task RL agent, one could perform some form of advantage-weighted regression [1] for skill acquisition. In this setting, one would weigh offline trajectories, over which to learn skills, based on their optimality. We have added these discussion points to Sections 4.1.
> >
> > [1] Peng, Xue Bin, et al. "Advantage-weighted regression: Simple and scalable off-policy reinforcement learning." arXiv preprint arXiv:1910.00177 (2019).
> >
> > **It seems unfair that we compare against methods that do not leverage expert trajectories.**
> >
> > Only Hier-RecSAC and RecSAC do not leverage expert trajectories for pre-training. It is common in skill transfer papers to compare against agents trained from scratch directly on the transfer task, to gauge the importance of skill transfer (see Tirumala 2019, 2020, Perstch 2020, Galashov 2019, Wulfmeier 2020, Bagatella 2022).
> >
> > We additionally compare against 5 other recent peer reviewed methods that have access to the same expert trajectories for pre-training as a fair comparison gauging the importance of APES.
> >
> > **Experiment settings seem more complex than they need to be.**
> >
> > Our experimental setup (seeds and complexity) are similar to many related works. Our experimental setup is similar to the baselines we compare against (particularly Tirumala 2019, 2020), and we compare on sequential navigation (CorridorMaze) and manipulation (Stack) environments like Tirumala 2019, 2020, Perstch 2020, Galashov 2019, Wulfmeier 2020, Bagatella 2022.
> >
> > **Transferability may depend on more things than covariate shift.**
> >
> > As with many skill transfer works, we made the explicit assumption (see the final 3 sentences of Section 4.1) that behaviors present over source domains, when recomposed, suffice for solving the transfer domains. This has been referred to as the modularity and compositional generalization assumptions (see [2] for a great literature review, especially Section 5.2.1). APES is suited for this setting.
> >
> > While the modularity constraint may appear restrictive, the increasingly diverse the source domains are (arguably as desired in many settings such as lifelong learning [2] or offline RL), the increasingly probable that the optimal transfer policy can be obtained by recomposing a subset of the source abstractions. If this assumption does not hold, one could either fine-tune the existing skills during transfer, which would require tackling the catastrophic forgetting of skills [3], or train additional new skills. One could use exploration bonuses, e.g. [4], to guide learning of new and diverse skills. We leave this as future work, but have added a condensed version of this discussion to Section 4.1.
> >
> > [2] Khetarpal, Khimya, et al. "Towards continual reinforcement learning: A review and perspectives." arXiv preprint arXiv:2012.13490 (2020).
> >
> > [3] Kirkpatrick, James, et al. "Overcoming catastrophic forgetting in neural networks." Proceedings of the national academy of sciences 114.13 (2017): 3521-3526.
> >
> > [4] Eysenbach, Benjamin, et al. "Diversity is all you need: Learning skills without a reward function." arXiv preprint arXiv:1802.06070 (2018).
> >
> > **Several paper claims are incorrect or not well supported.**
> >
> > Assuming that the incorrect or unsupported claims refer to the previous points, we have modified the paper to address these concerns. If there are some other incorrect or unsupported claims, could the reviewer point them out and we would happily address them.

---

> > > ### Comment · Reviewer_pBgA · 2022-11-17
> > > **Response to Author rebuttal**
> > >
> > > I would like to thank the authors for their detailed rebuttal.
> > >
> > > The rebuttal has answered some of my concerns, which is appreciated. However, many times in the rebuttal the justification for some of the elements in the paper is because other researchers have done so. In particular, in terms of the number of seeds and reliability of the experiments, this is extremely problematic. The community is suffering from severe problems of reproducibility and we can't keep going forward with our old ways, and for that reason I have very little trust that the proposed approach actually helps. For example, APES-H1 is very close in terms of performance on pretty much all the task, and it is not clear if the additional complexity of learning *m* is justified, let alone deeply understood. Under the current evaluation there is no statistical difference between that baseline and the proposed approach on the more complex environment, and I wonder if that would also be the case for all domains if more seeds were ran.
> > >
> > > In terms of choices of the experiments, it seems like a lot of taken for granted in knowing what the assumptions and details were in the baselines compared against. This does not help the clarity of the paper and the experiments.
> > >
> > > Concerning the relation between covariate shift and transfer, it may be possible that some papers have shown improvements in transfer when considering IA, but there are many more papers that have not considered IA and still show significant transfer capabilities. The language in the paper has been improved but it still seems to make claims that are too strong. If we take the argument about covariate shift to the limit, we could condition on nothing, but would the agent be able to transfer? Of course this is extreme, but it is to show that the two concepts are not directly related. I feel like a more nuanced conversation is still needed.

---

> > > > ### Author Response · Authors · 2022-11-18
> > > > **Rebuttal to Reviewer's Response (Part 1)**
> > > >
> > > > We appreciate that the reviewer values several of the paper modifications which we believe has improved the paper quality thanks to their feedback. We continue by trying to address the remaining concerns.
> > > >
> > > > **APES-H1 is very close in performance to APES and it is not clear if the additional complexity of APES is justified or understood.**
> > > >
> > > > If we inspect APES’ attention masks (Figure 5), we see that the input information it attends (previous action(s), and current state - for CorridorMaze) is closest to APES-H1. This qualitative analysis provides us with an understanding as to why both perform similarly. More generally, without APES, one would need to either manually craft the attention mask requiring expensive domain knowledge or perform computationally demanding IA sweeps that require expensive evaluation on the transfer domains, defeating the purpose of sample-efficient transfer. _Learning_ the attention mask from data could perhaps produce better masks that can ever be hand-crafted and significantly reduce the computational burden when compared to IA sweeps. We have seen similar phenomenons happening again and again in many fields of ML (e.g. hand-crafted vs. learned filters in computer vision,  meta-learning RL algorithms vs hand-crafted ones). We provide a principled method for learning the attention masks, based on our expressivity-transferability theorems, and our experiments provide evidence that it is able to recover performance similar to ones that are discovered through exhaustive IA sweeps whilst avoiding manually crafted masks.
> > > >
> > > > APES’ objective, based on our expressivity-transferability theorems, balances a trade-off between being able to represent skills present over the source domains (Theorem 3.2), and minimizing information conditioning and thus favoring generalization (Theorem 3.1). This is empirically backed by Table 3 (exhibiting almost zero variance, providing confidence in results), showing that APES achieves the best trade-off, with the highest distillation (skill representation) to entropy (information conditioning) ratio out of all methods. Our theorems and analysis allow us to understand APES’ success, and justify the formulation of the method. Our qualitative analysis further provides insights and understanding as to why our method succeeds. Figure 5 provides further evidence, beyond our 4 seeds, as to why APES is closer in performance to APES-H1 than other methods. Figures 3 and 6 provide insights as to why methods that lack priors or hierarchy fail. Appendix E.1 explains why for our domains, APES’ learnt attention is favorable for transfer.
> > > >
> > > > **Not sure that if more seeds were run that APES would outperform APES-H1.**
> > > >
> > > > An important contribution of APES is its ability to discover a high performing IA in an automated, domain-dependent, manner. Even if APES-H1 were the optimal choice, this is domain dependent, and would require exhaustive IA sweeps (on the transfer domains) to discover or expensive domain expertise. APES automates this process based on our expressivity-transferability theorems and does not need expensive transfer domain evaluation for choosing IA. In addition, we’d like to add that our APES-H1 baseline was introduced by Bagatella 2022, concurrent with our work. Compared to all other baselines that came out before APES, the performance gap with APES is significantly larger. In addition, whilst APES-H1 may be close to an optimal IA for our domains, this may not be the case for other domains. A method that automates this choice in a domain dependent manner is of importance for the skill-transfer field.
> > > >
> > > > **Multiple papers show significant transfer capabilities without considering IA.**
> > > >
> > > > We do not contest that effective transfer can be achieved without explicitly considering/exploring IA choices, as is the case for Pertsch 2020 on the domains they explore. However, we show, both theoretically and empirically, that considering IA can provide considerable performance gains. For example, for all our domains APES-H20 is the method with no IA between the hierarchical policy and prior (they are both conditioned on the history of length 20). In this setting, we see considerable performance gains by introducing IA and conditioning the prior on less information. APES automates this choice in a data-driven, domain-dependent, manner. The analysis in Section 5.4 and Appendix E.2, specifically Figure 4 and Table 8, demonstrates that APES-H20 performs poorly due to covariate shift between domains. In addition, Table 3 shows that APES obtains a far higher skill distillation to information conditioning ratio than APES-H20, which coupled with Theorem 3.1 and 3.2 explains its relative success.

---

> > > > > ### Author Response · Authors · 2022-11-18
> > > > > **Rebuttal to Reviewer's Response (Part 2)**
> > > > >
> > > > > **If we take the covariate shift argument to the limit and condition on nothing then the agent may not transfer favorably.**
> > > > >
> > > > > We would like to highlight that we do not claim that covariate shift alone influences transferability and we thus do not promote zero information conditioning. Our paper provides theory and empirical evidence behind the expressivity-transferability trade-off of skills across domains, for hierarchical KL-regularized RL. For hierarchical KL-regularized RL, this trade-off is dependent on IA choice.
> > > > >
> > > > > We point the reviewer to both Theorems 3.1 and 3.2, rather than just Theorem 3.1 that relates to covariate shift. Theorem 3.2 states that by conditioning on less information, we distill less expressive behaviors into our prior, thus also hurting transferability. As such, both theorems combined lead to the expressivity-transferability trade-off that we discuss in the paper, and is dependent on IA choice. Conditioning on too much information hurts transferability due to increased covariate shift. Conditioning on too little also hurts transferability due to reduced prior expressivity. Therefore, there is a trade-off between conditioning on too much or too little information. An analogy would be regularization techniques in supervised learning to prevent overfitting and improve transferability: there exists a sweet spot between regularizing too heavily and being unable to represent the underlying function, and regularizing too little and overfitting. APES automates this choice by encouraging a low level of information conditioning that is still able to adequately represent the behaviors present over the source domains.
> > > > >
> > > > > **The authors assume the readers know the assumptions and details that the baselines took. This does not help paper clarity and the experiments.**
> > > > >
> > > > > The skill transfer baselines make similar assumptions to us, which is why we compare against them. They all implicitly make the compositional generalization assumption [1], that behaviors present over source domains, when recombined, can solve the transfer tasks. As such, they all assume multi-task source domains for skill discovery. They additionally compare on similar source and transfer domains to us, as mentioned in our previous rebuttal and highlighted in the modified paper, further demonstrating that they are targeting a similar problem. We and our baselines assume that dynamics between domains are constant and only reward functions differ. We do not believe that we, or our baselines, make any further major assumptions. Could the reviewer point us to which further assumptions they would like us to highlight in the paper?
> > > > >
> > > > > We would also like to emphasize that in our paper, APES and all baselines are trained with the same training regime (same method for learning the critic and policy - see Section 4) and architecture (Section 3.0). As such, the only factors of variation are those in hierarchical and IA choices, allowing us to really understand in a clear manner how they influence performance. We provide full experimental setups in Appendix D, and in Table 6 visually depict all factors of variation across baselines for clarity.
> > > > >
> > > > > [1] Khetarpal, Khimya, et al. "Towards continual reinforcement learning: A review and perspectives." arXiv preprint arXiv:2012.13490 (2020).
> > > > >
> > > > > **Certain paper claims are too strong and a more nuanced conversation is needed.**
> > > > >
> > > > > May the reviewer point us to which statements are too strong and where more nuanced conversation is needed? As per our previous discussion, we modified the paper to highlight that our expressivity-transferability statements (and in turn our covariate shift ones) are in the context of hierarchical KL-regularized RL. We have also modified the paper to clarify our assumptions on source and transfer domain similarity: that the source domains exhibit the behaviors (skills) necessary to solve the transfer domains (see Section 4.1 and our previous rebuttal). If the reviewer can point us to the exact statements they have an issue with we will happily correct them.

---

> > > > > > ### Author Response · Authors · 2022-11-18
> > > > > > **Rebuttal to Reviewer's Response (Part 3)**
> > > > > >
> > > > > > **Without more seeds one cannot trust that the proposed approach actually helps.**
> > > > > >
> > > > > > As mentioned in our other responses, we believe that our theorems (Theorems 3.1 and 3.2) and qualitative analysis (Figures 3, 5, 6) provide insights that explain the quantitative empirical results. Theorem 3.1 and 3.2 explain the expressivity-transferability trends seen in Table 2. Table 3, coupled with Theorems 3.1 and 3.2, further demonstrates that APES achieves the best expressivity-transferability trade-off. Theorem 3.1 explains the trends in Figure 4. Figure 5 provides insight as to why APES is closest in performance to APES-H1. Figures 3 and 6, coupled with the qualitative analysis in Appendix E, show which skills APES learns (block stacking or corridor traversals) and hence why it transfers so favorably. In fact, this is why reviewer ia3Q values our domains, as they “are insightful because the skills in these cases are intuitive”, hence easing analysis. We note that the only approach that is close to APES’ performance is APES-H1, work done concurrent to our research. Crucially, unlike all other approaches (including APES-H1), APES automates their IA choice in a domain-dependent manner, bypassing arduous IA sweeps (requiring multiple expensive evaluations on the transfer domain that combined reduce the sample-efficacy) and hand-crafted masks. We believe that this automation is APES’ main value, and our theory and many analyses provide further justifications and understanding behind the method. We will modify the paper to make these points about APES’ value clearer.
> > > > > >
> > > > > > We can run additional seeds if desired, but due to computational constraints, these results will not be ready before the end of the rebuttal period.
> > > > > >
> > > > > > **In general**
> > > > > >
> > > > > > We are happy to amend the paper to incorporate these discussions and clarifications if the reviewer desires. We believe this could further improve paper accessibility.

---

### Official Review · Reviewer_RbDM · 2022-10-25

**Confidence:** 3
**Correctness:** 3
**Technical Novelty And Significance:** 3
**Empirical Novelty And Significance:** 2
**Recommendation:** 6

**Clarity, Quality, Novelty And Reproducibility:**

- The presentation is basically clear, but there are some points for improvement (please see my comment above).
- The quality could be improved by strengthening the theoretical or empirical connection between the presented theorems and the proposed method.
- While the masking of input variables is not uncommon, the proposition of the masking method for the KL-regularized RL based on the theorems with KL divergence has some novelty.

**Strength And Weaknesses:**

Strengths

- I like the presentation of the expressivity-transferability trade-off using the theorems in terms of KL divergence, and the attempt to connect them with the proposed method.
- They analyze the experimental results in multiple ways, which could help understanding the empirical effectiveness of the proposed approach.

Weaknesses

- I'm concerned that by using the soft masks to allow them to be trained, the method could lose most of its theoretical connection with Thm.3.2 and thus the expressivity-transferability trade-off. It is because, unless some mask is strictly zero (or near-zero in practice) the information the corresponding variable carries does not go away. This is probably supported by Fig.5, where the minimum mask value is $\approx \exp(-3.1) \approx 0.045$. Their corresponding variables may not play big roles in the prediction, but it's hard to say that they are completely ignored.
- The scales of the soft masks can largely be affected by the scales of the input variables. I'm curious if they are properly normalized in the experiments.
- The writing could be improved in some ways. In Eq.3, $\pi$ and $\pi_0$ are not conditioned on the masks and $\mathcal{O}_{\text{APES}}$ does not take the masks as its input. The boldfacing of the inputs in the text and Thm.3.2 do not match. The second sentence of Sec.2 is missing an "and".

**Summary Of The Paper:**

The authors propose an approach to the problem of transferring learned behaviors between tasks in a sequential setting, called Attentive Priors for Expressive and Transferable Skills (APES). They first present two theorems: a) with more input variables, the shift of the input distributions in terms of KL divergence gets bigger and b) conditioning on more variables leads to better expected knowledge distribution in terms of the KL divergence between output distributions, which together suggest an expressivity-transferability trade-off. The authors then suggest APES, which learns soft masks for input to each module in the KL-regularized hierarchical policies. They test and perform analyses of their approach in the CorridorMaze and Stack environments to show the empirical effectiveness of the approach.

**Summary Of The Review:**

I find the connection between the theorems on the expressivity-transferability trade-off and the proposed method somewhat interesting, but my major concern is that the connection may not hold in the case of the soft mask learning.

---

> ### Author Response · Authors · 2022-11-12
> **Reviewer RbDM Rebuttal**
>
> **Because soft attention is employed the authors lose connection to theory, which is supported by attention values never below 0.045 (Fig 5).**
>
> We clarify to the reviewer that Fig 5 plots marginalized log_10(m), aggregating attention across intra-observation/action dimensions. We also highlight that in Fig 5 and 7 we report log base 10 rather than the natural logarithm. We plot full attention maps, across individual observation-action dimensions in Fig 7 in Appendix E.1. In Fig 7, attention values exhibit a wide range, from 10^-6 to 10^-0.5, 5.5 orders of magnitude greater range than what the reviewer thought. As such, we empirically demonstrate that soft attention leads to many attention values approaching 0, similar to Salter 2020 and [1], thus closely aligning with the theory. Tables 2, 3, and Figure 4 empirically demonstrate that APES outperforms previous methods by discovering the most favorable expressivity-transferability trade-off.
>
> We have modified Section 4 and 5.5 to clarify that whilst soft attention does not strictly eliminate information in the same way that hard attention does, in practice many elements tend towards 0. In addition, we clarify in Fig 5 and 7 that we use log base 10.
>
> [1] Mott, Alexander, et al. "Towards interpretable reinforcement learning using attention augmented agents." Advances in Neural Information Processing Systems 32 (2019).
>
> **The scales of input variables together with attention values heavily influence the information available to the policy. Are input scales normalized?**
>
> In our domains, the scales and ranges of the observation-action dimensions do not vary drastically (with observations corresponding to cartesian positions and have a similar magnitude to the actions). However, if they did, one could either normalize each dimension by the given range if available, or by a running mean and std, akin to DDPG.
>
> **The masking of input variables is not uncommon.**
>
> Whilst many hierarchical and KL-regularized works employ input masking, as reviewer ia3Q points out, we present a unifying perspective connecting skill-learning algorithms that condition on different aspects of the observation-action history spaces, and analyze them under a common umbrella. To our knowledge, we are the first to learn this masking in a data-driven manner.
>
> **Technical and empirical novelties are marginal.**
>
> We compare against many recent, high-performing, widely recognised baselines (e.g. Pertsch at Conference of Robot Learning 2020, Galashov at the International Conference of Learning Representations 2019, Wulfmeier at Robots Science and Systems 2020, Tirumala at the Journal of Machine Learning Research 2022). APES outperforms across various environments and reward sparsity levels.
>
> **Eq.3 should condition on m.**
>
> We thank the reviewer for pointing this out. We have made the necessary modifications to Eq. 3 and Eq. 4.
>
> **Boldfacing of input variables in text do not match Thm 3.2.**
>
> We thank the reviewer for spotting this error. We have corrected this inconsistency.
>
> **Second sentence in Sec 2 is missing an ‘and’.**
>
> We have added the ‘and’.
>
> **Several paper’s claims are incorrect or not supported.**
>
> Assuming that the incorrect or unsupported claims refer to the previous points, we have modified the paper to address these concerns. If there are some other incorrect or unsupported claims, could the reviewer point them out and we would happily address them.

---

> > ### Comment · Reviewer_RbDM · 2022-11-13
> > **Response to Authors**
> >
> > Thanks for the detailed response from the authors.
> >
> > I appreciate that the updated manuscript clarifies the theoretical and empirical aspects of the use of the soft attention and that the logarithm with base 10 is used in the figures. Also, I agree that classifying this work's technical novelty as only marginal was a bit harsh, considering the main goal of this work. I am adjusting my score to 6.

---

> > > ### Author Response · Authors · 2022-11-13
> > > **Response to Reviewer RbDM**
> > >
> > > We are glad that the reviewer appreciates the changes made to the manuscript and values the technical novelty of the paper. We thank the reviewer for their considerate feedback which has helped strengthen the paper quality.

---

### Official Review · Reviewer_Dj5g · 2022-10-25

**Confidence:** 3
**Correctness:** 3
**Technical Novelty And Significance:** 3
**Empirical Novelty And Significance:** 3
**Recommendation:** 6

**Clarity, Quality, Novelty And Reproducibility:**

Clarity:
- The write-up makes liberal use of italics, which hurts readability as the reader's focus is constantly diverted. I'd suggest to use italics much more sparingly and reserve it for crucial results or insights (e.g., re-formulating the Theorems in 3.1. seems fine, but the highlights in the first paragraph of Section 4 are not helpful).
- I'm unclear on how $m$ is learned exactly; Section 4 could be improved in this regard. For example, from Eq. (3) it seems like the only thing that gets optimized is its entropy?
- The paper initially talks about sequential tasks (e.g., in 3.1), but the experiments are carried out in a transfer setting where experts for less complex task instantiations are available for initialization via BC.
- To prevent misunderstandings, it would be good to include the tube around the blocks in the environment rendering in Fig. 2(b)
- When defining losses, I would strongly suggest using a more widely recognized symbol, i.e., $\mathcal{L}$ instead of $\mathcal{O}$
- The introduction promises experiments over a "wide range of domains", yet only 3 tasks in 2 domains are considered.

Novelty:
- Highlighting and addressing a fundamental trade-off
- For the related work, consider Gehring et al., 2021 (https://arxiv.org/abs/2110.10809) which address a related trade-off in the context of unsupervised skill discovery and likewise propose a data-driven solution.
- The overall setup seems to largely follow Tirumala et al. (2020), with the addition of a softmax over past states

Reproducibility:
- Source code is not provided, but the authors provide their algorithm and hyper-parameters in the Appendix. For full reproducibility, more information about (or access to) the scripted expert policies would be required.

**Strength And Weaknesses:**

Strengths:
- Highly relevant topic with focus on a fundamental trade-off when distilling behavior priors.
- Well-motivated method that effectively automates this trade-off as part of policy training.
- Strong performance in the tasks considered.

Weaknesses:
- Information Asymmetry not only concerns the amount of history presented, but also the dimensions of the state that are relevant. It would have been nice to see a further application of attention across state dimensions, e.g., in a simple navigation task. This could then also involve environments for which the selected baselines have been originally developed.
- On a similar note, the considered environments are relatively simple, although the sparse reward structure renders them of course challenging for an RL algorithm. It would be great to see that the proposed method does not introduce regressions on tasks in which standard behavior priors work well.
- The presentation and writing is at times unclear and could be improved (see below).

**Summary Of The Paper:**

The submission addresses the problem of information asymmetry (IA) in hierarchical behavior priors as proposed in Tirumala et al. (2020). The role of IA is to limit the prior's access to observations or actions so that the behaviors that are captured are general rather than environment- or instance-specific (for example, restricting observations to proprioceptive sensing for simulated robots might result in the prior capturing general locomotion behavior). As such, the concrete choice of IA is a design decision usually performed by humans, and the main contribution of the paper here is to instead learn this asymmetry in a data-driven manner. The setup that is examined here is that of a high-level policy prior, and IA concerns the amount of past states presented to this prior. The proposed method employs soft attention on the entire history of states for learning IA.

**Summary Of The Review:**

This appears to be a good paper addressing a relevant topic. Due to issues wrt clarity and the limited experimental section, I'm leaning slightly in favor of rejection at this point.

---

> ### Author Response · Authors · 2022-11-12
> **Reviewer Dj5g Rebuttal (Part 1)**
>
> **It would be nice to see intra-observation attention which is then more applicable to environments for which alternate methods were developed.**
>
> We would like to clarify that APES does learn intra-observation and intra-action attention masks (m) across the history of observations and actions (with m spanning individual observation-action dimensions). We analyze intra-observation/action attention in Fig 7 and Appendix E.1. APES ignores redundant aspects of the observation-action spaces, such as the gripper state for Stack, thus minimizing information conditioning and improving transferability. We have modified Section 4 and 5.5 to clarify this, moving some of the Appendix analysis to the main paper.
>
> We point out that our environments are similar to those that our baselines were developed for and tested against. Similar to CorridorMaze, Tirumala 2019, 2020 and Petsch 2020 compare against sequential goal reaching domains (such as mazes). Like Stack, Tirumala 2019, 2020, Petsch 2020 and Wulfmeier 2019 evaluate on block manipulation environments. As such, we believe that the comparison with these methods is fitting and have modified Section 5 to highlight this.
>
> **Simple environments made difficult by reward structure. Would be nice to see if APES favors tasks which standard behavior priors were developed for.**
>
> We note that our environments are of similar (if not harder) complexity to other skill papers (e.g. Tirumala 2029, 2020, Pertsch 2020, Wulfmeier 2019). We compare APES against environments with varying reward sparsity levels (CorridorMaze’s sparse 2-corr and s-sparse 4 corr tasks) to see whether APES is sensitive to this setting. APES consistently outperforms all baselines, although the degree depends on sparsity level.
>
> **Reviewers talk about sequential task setting but assume expert behaviors over source tasks and train via BC.**
>
> We investigate how hierarchy, priors, and IA influence distillation and skill transfer across sequential tasks. As is common in sequential task learning, we assume the source tasks are solved before moving onto the transfer tasks. Therefore, for analysis purposes, it does not matter whether the source tasks are solved, and demonstrations provided, by a hardcoded expert or fully optimized RL agent. In-fact, many skill transfer methods, such as Pertsch 2021 and our baseline Pertsch 2020, also assume “expert” source task trajectories for skill discovery. If one wished to learn skills from sub-optimal offline trajectories, such as those from the replay buffer of a lifelong learning sequential task RL agent, one could perform some form of advantage-weighted regression [1] for skill acquisition. In this setting, one would weigh offline trajectories, over which to learn skills, based on their optimality. We have added these discussion points to Sections 4.1.
>
> [1] Peng, Xue Bin, et al. "Advantage-weighted regression: Simple and scalable off-policy reinforcement learning." arXiv preprint arXiv:1910.00177 (2019).
>
> **It is unclear how m is optimized in Eq 3. Is entropy the only thing optimized for?**
>
> We have reworded Section 4 and Eq 3 to clarify this point. To summarize: m is the attention mask over all observation-action dimensions in the history. For the prior, m influences its ability to distill policy behavior (the KL-divergence loss in Eq 3). Therefore, both loss components influence m, encouraging minimal information conditioning (a low-entropic mask) that pays attention to relevant history dimensions (to minimize the KL loss).
>
> **Please add expert policy setup to appendix for reproducibility.**
>
> We have added this to Appendix C.
>
> **Authors promise a wide range of domains yet only compare against 2.**
>
> We have reworded the introduction and conclusion to explain that we investigate over various reward sparsity levels and domains. We compare against a similar range of environments to related skill transfer papers, and believe that our theory, coupled with empirical analysis over various extrapolation and reward sparsity levels, provide valuable insights to the community.
>
> **Use more widely recognised symbols for losses (L rather than O).**
>
> We thank the reviewer for this suggestion. We have changed the necessary equations as requested.
>
> **To prevent misunderstanding it would be good to include a tube around blocks in Fig 2(b).**
>
> We cannot currently make this modification, but assure the reviewer that we will include this for the camera ready version.
>
> **Liberal use for italics which hurts readability.**
>
> We agree with the reviewer. As requested, we have reduced the use of italics in the paper.

---

> > ### Author Response · Authors · 2022-11-12
> > **Reviewer Dj5g Rebuttal (Part 2)**
> >
> > **Add Gehring et al 2021 to related works.**
> >
> > We thank the reviewer for sharing this interesting and related paper with us. Gehring 2021 introduces a hierarchical framework that learns multiple skills of varying levels of expressivity, leading to a combinatorial action space. Works like Pertsch 2020, 2021 demonstrate that when the number of skills grows, that priors can be essential for narrowing exploration over the skill space. Extending this work with learnt skill priors, using APES to ensure they transfer favorably, is an interesting future direction. We have added this discussion to the related works section.

---

> > ### Comment · Reviewer_Dj5g · 2022-11-14
> > **Thank you for the updates**
> >
> > Thank you for the updates and clarifications! Given these, I'm raising my score to 6.
> >
> > Regarding environments, Tirumala 2019 also perform experiments in environments with more complex dynamics, i.e., with Ant and Humanoid robots, where locomotion and goal/object interaction can be separated in terms of behavior. Figure 7 in your submission shows the desired effect (excluding gripper state), but showing that your method scales up to higher-dimensional states would be very interesting.

---

> > > ### Author Response · Authors · 2022-11-14
> > > **Response to Reviewer Dj5g**
> > >
> > > We are glad that the reviewer appreciates the changes made to the manuscript. We also agree that extending APES to higher-dimensional observation-action domains, such as image-based RL, is an exciting future direction which we plan to explore in the future.

---

### Author Response · Authors · 2022-11-12
**Joint Rebuttal**

We thank the reviewers for their detailed and insightful feedback, which we believe has greatly improved the paper quality. We are glad that the reviewers value: our theorems and their importance to the highly relevant topic of skill transfer; our well motivated method unifying existing model-free skill-learning methods under a common umbrella; our broad and insightful analysis with APES compared to many relevant baselines; our strong performance. The main concerns are with respect to written clarity, leading to a couple of paper misinterpretations. Specifically, reviewer Dj5g was not aware that our attention masks cover intra-observation/action attention, which was one of their concerns. Reviewer RbDM misinterpreted Figure 5, due to a similar misunderstanding, leading to concerns between the practical relation between APES and our theorems. To address this, we have made numerous modifications to the manuscript which we believe have greatly improved the paper clarity. To ease the second round of reviews, we have highlighted all the modifications by red text.

**To summarize, we have modified:**

**Section 1 (Introduction)** - We provide more IA examples and intuitively define the expressivity-transferability trade-off early on to help the reader understand the context of the paper.

**Section 3.1 (The IA Expressivity-Transferability Trade-Off)** - We clarify the context of the IA trade-off - skill transfer 	using hierarchical KL-regularized RL - and highlight the differences between the multi- and sequential- task settings.

**Section 4 (APES)** - We clarify APES’ objective, how it influences attention, and how it relates to the expressivity-transferability theorems despite soft-attention being used rather than hard-attention.

**Section 4.1 (Training Regime and IA setup)** - We explain why we assume expert behaviors over source domains, and the assumptions that we make about source and transfer domains and how they influence transferability.

**Section 5 (Experiments and Results)** - We justify how our baselines are well suited for APES and our domains.

**Section 5.5 (APES’ Results)** - We clarify that we learn intra-observation/action attention and analyze these results. We also explain Figure 5 and how most attention values tend to 0 thus aligning closely with our theorems.

**Section 5.7 (Skill-Level Exploration Analysis)** - We explain in more detail the exploration analysis for the Stack environment.

**Section 6 (Related Work)** - We add suggested missing related works, including some options frameworks and model-based RL methods.

**Appendix C** - We add the expert setup for the source domains for completeness.

We made other modifications, as highlighted in the individual reviewer rebuttal responses, which we omit naming here for conciseness.

---

### Decision · Program_Chairs · 2023-01-20

**Decision:**

Accept: poster

**Justification For Why Not Higher Score:**

Not enough enthusiasms from reviewers (two 6 and one 8; one 5).

**Justification For Why Not Lower Score:**

It has novel contributions that can benefit the community.

**Metareview: Summary, Strengths And Weaknesses:**

The paper considers the sequential transfer in RL and proposes to combine two approaches for better transfer: two-level hierarchical RL and KL-regularized RL with a learned prior. It argues that the use of Information Asymmetry, in which different modules of the method have access to different information about the problem, is a key to learn transferrable hierarchy and priors. As opposed to previous work, it learns how to create the Information Asymmetry by learning a masking function.

The paper has some theoretical justification showing a tradeoff between expressivity and transferability. It also performs empirical studies.

Three out of four reviewers are positive about this paper, and one is on the weak negative side. They found the unifying framework and the expressivity-transferability trade-off interesting (Reviewer ia3Q, RbDM), the automating the trade-off well-motivated (Reviewer Dj5g), and the experimental results convincing (Reviewers ia3QH, Dj5g).

There are some concerns though such as the unclear presentation at points (especially in the theory part), not a strong connection between the theoretical parts and the empirical ones, and the use of small number of seeds (4) in the empirical part.

Several of these concerns have been addressed in the revised version of the paper, and most reviewers are satisfied. There were long discussions between Reviewer pBgA and the authors, which did not end in a final agreement.

After reading the paper myself, I have some suggestions:

- The number of seeds is four (4). This is small. Although this choice is aligned with the current practice of a subset of the deep RL community, I agree with Reviewer pBgA that bad practice of others is not a good justification for its continuation. The paper does not provide a statistical significant test on the results.

Looking at Table 2, I see that there is relatively good gap between most results, considering the (inaccurate estimate) of the standard deviations, perhaps except for APES vs APES-H1 in 4 block domain. Therefore, I am not too worried about the validity of the results, but one cannot be completely confident given this small number of seeds and the lack of a formal hypothesis test.

I strongly recommend the authors to re-run their experiments with more seeds. My request should be interpreted as that I trust authors to do this change in their final version. If this was a journal paper, I would recommend a major revision with this as one of the main requests.

- APES happens to find a masking that makes it similar to APES-H1 in the tested domains, as shown in Figure 5. The authors rightfully mention that this is domain-dependent and APES' benefit is that it automates the process of finding the right masking.

Is there any domain that APES finds a less trivial masking that doesn't focus on the most recent past? Having such a result would make the paper's argument on the importance of automating the process of finding the mask more convincing. This is a suggestion to make the paper stronger, and is not a requirement.

- The experiments in Table 2 look at APES-Hk with k being in the set of 1, 10, and 20 (as well as APES-S, which seems to be equivalent to k=0). It would be insightful to have a wider range of ks and present them in a graph. This provides a clear visualization of the tradeoff compared to showing only 4 points.

Overall,  this is a reasonably good paper. As most reviewers agree that this should be published, I would also recommend its acceptance with the understanding that the authors would improve their experiments.

**Note From Pc:**

if the above contains the word "oral" or "spotlight" please see: "oral" presentation means -> notable-top-5% and "spotlight" means -> notable-top-25%. As stated in our emails, we are disassociating presentation type from AC recommendations